# Common variants in Alzheimer's disease and risk stratification by polygenic risk scores

Genetic discoveries of Alzheimer's disease are the drivers of our understanding, and together with polygenetic risk stratification can contribute towards planning of feasible and efficient preventive and curative clinical trials. We first perform a large genetic association study by merging all available case-control datasets and by-proxy study results (discovery $n = 409,435$ and validation size $n = 58,190$). Here, we add six variants associated with Alzheimer's disease risk (near *APP, CHRNE, PRKD3/NDUFAF7, PLCG2* and two exonic variants in the *SHARPIN* gene). Assessment of the polygenic risk score and stratifying by *APOE* reveal a 4 to 5.5 years difference in median age at onset of Alzheimer's disease patients in *APOE* ε4 carriers. Because of this study, the underlying mechanisms of *APP* can be studied to refine the amyloid cascade and the polygenic risk score provides a tool to select individuals at high risk of Alzheimer's disease.

Thus far, multiple loci associated with Alzheimer's disease (AD) have been described next to causal mutations in two subunits of γ-secretases, membrane-embedded aspartyl complexes (*PSEN1, PSEN2 genes*), and the gene encoding one target protein of these proteases, the amyloid precursor protein gene *(APP)*. The most prominent locus, *APOE*, was detected almost 30 years ago using linkage techniques[1]. In addition, genome-wide association studies (GWAS) of AD case-control datasets and by-proxy AD case-control studies have identified 30 genomic loci that modify the risk of AD[2–7]. These signals account for ~31% of the genetic variance of AD, leaving most of the genetic risk as yet uncharacterized[8]. Further disentangling the genetic constellation of common genetic variations underlying AD can drive our biological insights of AD and can point toward novel drug targets.

There are over 50 million people living with dementia and the global cost of dementia is well above 1 trillion US$[9]. This means there is a medical and economical urgency to efficiently test interventions that are under development. Therefore, to increase power and reduce duration of trials, pre-symptomatic patients that are at high genetic risk of disease are increasingly developed[10]. However, only carriers of causal mutations (*APP, PSEN1,* and *PSEN2*) and the *APOE* ε4 allele are considered high risk, while other common and rare genetic variants are ignored[11]. Despite that, the combined effects of all currently known variants in a polygenic risk score (PRS) is associated with the conversion of mild cognitive impairment to AD[12,13], the neuropathological hallmarks of AD, age at onset (AAO) of disease[14–17] and lifetime risk of AD[18].

In this work we aim to comprehend and expand the knowledge of the genetic landscape underlying AD and provide additional evidence that a PRS of variants can be a robust tool to select high risk individuals with an earlier AAO. We first performed a meta-GWAS integrating all currently published GWAS case-control data, by-proxy case-control data, and the data from the Genome Research at Fundació ACE (GR@ACE) study[19]. We confirm the observed associations in a large independent replication study. Then, we construct an update of the PRS and test whether the effects of the PRS are influenced by diagnostic certainty, sex and AAO groups. Lastly, we test whether the PRS could be used to identify individuals at the highest odds of having AD and we compared AAO of the AD cases. This study describes the identification of six variants associated with AD risk and provides an extended PRS tool to select individuals at high risk of AD.

## Results

**Meta-GWAS of AD**. We combined data from three AD GWASs: the summary statistics calculated from the GR@ACE[19] case-control study (6331 AD cases and 6055 controls), the IGAP[20] case-control study (up to 30,344 AD cases and 52,427 controls) and the UKB AD-by-proxy case-control study[21] (27,696 cases of maternal AD with 260,980 controls, and 14,338 cases of paternal AD with 245,941 controls, Fig. 1, Supplementary Data 1). Although we observed inflation in the resulting summary statistics ($\lambda$ median = 1.08; see Supplementary Fig. 1d), it was not driven by an un-modeled population structure (LD score regression intercept = 1.036). The full details of the studies are described in methods. After study-specific variant filtering and quality-control procedures, we performed a fixed effects inverse-variance-weighted meta-analysis[22] on the summary statistics of the three studies. Using this strategy, we identified a genome-wide significant (GWS) association ($p < 5 \times 10^{-8}$) for 36 independent genetic variants in 35 genomic regions (the *APOE* region contains signals for ε4 and ε2). As a sensitivity analysis, we removed the AD-by-proxy study and compared the resulted effect estimates with and without this dataset. We found a high correlation between the effect estimates from the case-control and by-proxy

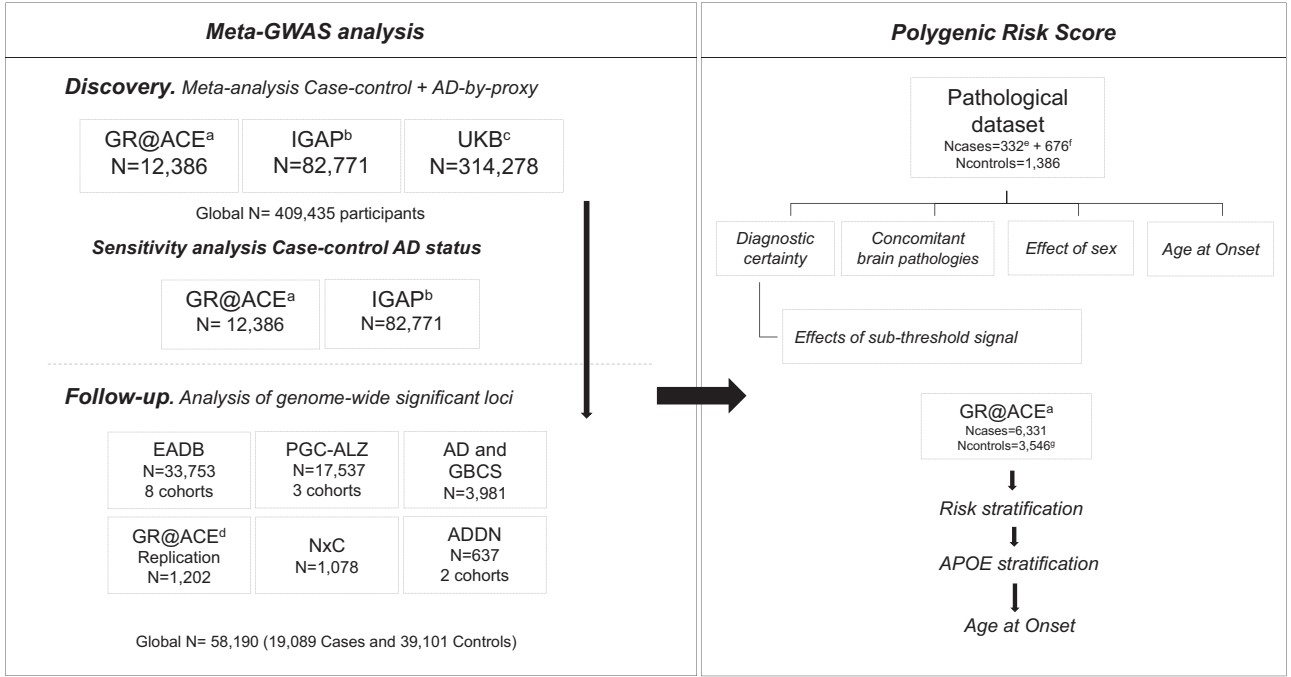

**Fig. 1 Flow chart of analysis steps.** Discovery meta-analysis in GR@ACE, IGAP stage 1 + 2 and UKBiobank followed by a replication in 16 independent cohorts. The genome-wide significant signals found in meta-GWAS were used to perform a Polygenic Risk Score in a clinical and pathological AD dataset. See Supplementary Methods to more information about the cohorts included and methods to the PRS generation. [a]Extended dataset (Moreno-Grau et al.[19]), [b]StageI + StageII (Kunkle et al.[20]), [c]By proxy AD: Meta-analysis of maternal and paternal history of dementia (Marioni et al.[21]), [d]Extra and independent GR@ACE dataset incorporated only for replication purposes, [e]Pathologically confirmed AD cases, [f]AD cases diagnosed based on clinical criteria, [g]Controls participants aged 55 years and younger. N = Total of individuals within specified data.

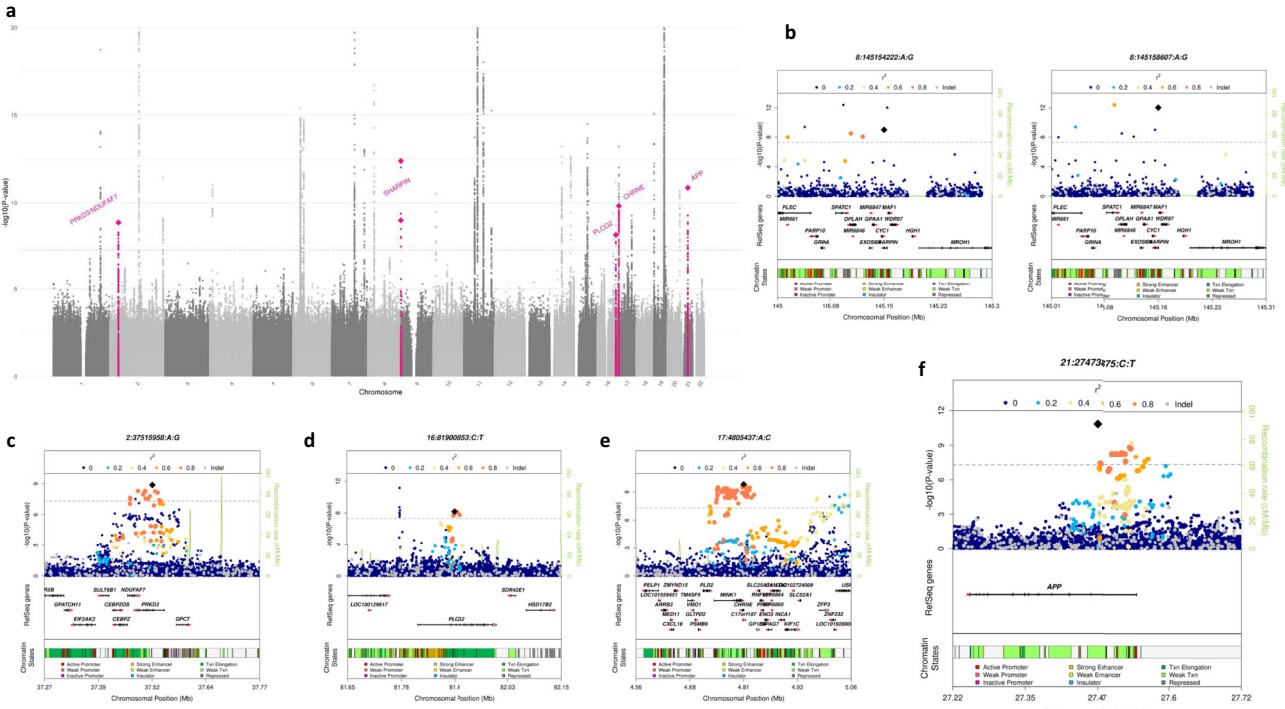

**Fig. 2 GWAS meta-analysis for AD risk ($N = 467,623$). a** Manhattan plot of overall meta-analysis for genome-wide association in Alzheimer's disease highlighting in pink the loci associated with AD in this study (*PRKD3/NDUFAF7*, *SHARPIN*, *CHRNE*, *PLCG2*, and *APP*). **b–f** Locus plots for the signals associated with AD in overall meta-analysis results.

approaches for the significant loci ($R^2 = 0.994$, $p = 8.1 \times 10^{-37}$; Supplementary Fig. 1e). Four genomic regions were not previously associated with AD (see Manhattan Plot, Fig. 2a).

Next, we aimed at replicating the associated loci in 16 cohorts (19,087 AD cases and 39,101 controls in total), many of them collected and analyzed by the European Alzheimer's Disease Biobank (JPND-EADB) project. We tested all variants with suggestive association ($p < 10^{-5}$) located within a 200 kb region from the sentinel SNP. Overall, 384 variants were tested in the replication datasets (Supplementary Data 2). Discovery and replication were combined, and we identified associations in six variants comprising five genomic loci annotated using FUMA[23] (Table 1, Fig. 2b–f, Supplementary Fig. 2 and Supplementary Results). In *APP*, we identified a common (MAF = 0.46) intronic variant associated with a reduced risk of AD (rs2154481, OR = 0.95 [0.94–0.96], $p = 1.39 \times 10^{-11}$, Fig. 2f). In *SHARPIN* (SHANK Associated RH Domain Interactor) gene, we found two missense mutations (rs34173062/p.Ser17Phe and rs34674752/p.Pro294Ser) that are in linkage equilibrium ($R^2 = 1.3 \times 10^{-6}$, $D' = 0.014$, $p = 0.96$). Both missense variants increased AD risk (p.Ser17Phe, MAF = 0.085, OR = 1.14 [1.10–1.18], $p = 9.6 \times 10^{-13}$ and p. Pro294Ser, MAF = 0.052, OR = 1.13 [1.09–1.18], $p = 1.0 \times 10^{-9}$, Fig. 2b). A variant close to the genes *PRKD3* and *NDUFAF7* (rs876461, MAF = 0.143) emerged as the most significant variant in the region after the combined analysis (OR = 1.07 [1.05–1.09], $p = 1.3 \times 10^{-9}$, Fig. 2c). In the 3'-UTR region of *CHRNE* (Cholinergic Receptor Nicotinic Epsilon Subunit), rs72835061 (MAF = 0.085) was associated with a 1.09-fold increased risk of AD (95% CI [1.06–1.11], $p = 1.5 \times 10^{-10}$, Fig. 2e). Our analysis also strengthened the evidence of association with AD for three additional genomic loci including an association with a variant in *PLCG2* (rs3935877, MAF = 0.13, OR = 0.92 [0.90–0.95], $p = 6.9 \times 10^{-9}$, Fig. 2d), and confirmed another common variant in *PLCG2*, a stop gain mutation in *IL-34* and a variant near *HS3ST1* (Table 1, Supplementary Fig. 3 and Supplementary Data 2, 3). We were not

able to replicate two loci (*ELK2AP* and *SPPL2A* regions) that showed suggestive association with AD ($p < 1 \times 10^{-7}$ in discovery).

**Polygenic risk scores.** In order to assess the robustness and combined effect of the genetic landscape of AD (Fig. 3, Supplementary Data 4), we constructed a weighted PRS based on the 39 genetic variants (excluding *APOE* genotypes) that showed GWS evidence of association with AD (see Methods, Fig. 4 and Supplementary Data 5). We tested if the association of the PRS with AD is independent of clinically important factors that are considered in the selection of individuals for clinical trials. First, we showed that the association of the PRS with clinically diagnosed AD cases is similar to the association with pathologically confirmed AD (OR = 1.30 vs. 1.38, per 1-SD increase in the PRS). In this setting, adding variants below the GWS threshold did not lead to a more significant association of the PRS with AD (Fig. 4a). Next, we tested whether the PRS was associated with AD in the presence of concomitant brain pathologies (besides AD). Among our autopsy-confirmed AD patients ($n = 332$), 84% had at least one concomitant pathology, and the PRS was associated with AD in the presence of all tested concomitant pathologies (Fig. 4b). Moreover, the patients often had more than one concomitant pathology (48.8%), but no difference was observed in the effect estimate of the PRS when more than one pathology was present (Fig. 4b). Last, we investigated the effect of sex and AAO (Fig. 4c). Our analysis revealed that the effect of the PRS was the same in both sexes (Fig. 4c) and was consistent with both early-onset (onset before 65 years; OR = 1.58, 95% CI [1.22–2.05], $p = 5.8 \times 10^{-4}$) as well as with late-onset AD (onset later than 85 years; OR = 1.29, 95% CI [1.10–1.51], $p = 1.5 \times 10^{-3}$).

PRSs has the potential to early identify subjects at risk of complex diseases[24]. To identify people at the highest genetic risk of AD based on the PRS, we used the validated 39-variants

**Table 1 Association for the AD loci selected for follow-up.**

| Chr | Closest gene | SNP | BP | A1 | A2 | Freq A1 | Discovery meta-analysis | | Follow-up datasets | | Overall | |
|---|---|---|---|---|---|---|---|---|---|---|---|---|
| | | | | | | | OR [CI 95%] | P | OR [CI 95%] | P | OR [CI 95%] | P |
| **Variants showing novel genome-wide significant association with AD** | | | | | | | | | | | | |
| 2 | PRKD3/NDUFAF7 | rs876461 | 37515958 | A | G | 0.143 | 1.07 [1.04–1.09] | $9.14 \times 10^{-7}$ | 1.08 [1.04–1.13] | $3.07 \times 10^{-4}$ | 1.07 [1.05–1.09] | $1.34 \times 10^{-9}$ |
| 8 | SHARPIN | rs34674752 | 145154222 | A | G | 0.052 | 1.11 [1.06–1.16] | $4.02 \times 10^{-6}$ | 1.20 [1.10–1.31] | $1.65 \times 10^{-5}$ | 1.13 [1.09–1.18] | $1.00 \times 10^{-9}$ |
| 8 | SHARPIN | rs34173062 | 145158607 | A | G | 0.085 | 1.16 [1.11–1.21] | $1.33 \times 10^{-11}$ | 1.09 [1.02–1.17] | $7.35 \times 10^{-3}$ | 1.14 [1.10–1.18] | $9.62 \times 10^{-13}$ |
| 16 | PLCG2 | rs3935877 | 81900853 | C | T | 0.868 | 0.92 [0.90–0.95] | $1.12 \times 10^{-7}$ | 0.92 [0.85–0.99] | $1.96 \times 10^{-2}$ | 0.92 [0.90–0.95] | $6.85 \times 10^{-9}$ |
| 17 | CHRNE | rs72835061 | 4805437 | A | C | 0.085 | 1.09 [1.06–1.12] | $3.92 \times 10^{-9}$ | 1.07 [1.02–1.12] | $7.83 \times 10^{-3}$ | 1.09 [1.06–1.11] | $1.51 \times 10^{-10}$ |
| 21 | APP | rs2154481 | 27473875 | C | T | 0.483 | 0.95 [0.93–0.96] | $9.26 \times 10^{-10}$ | 0.96 [0.93–0.99] | $3.31 \times 10^{-3}$ | 0.95 [0.94–0.96] | $1.39 \times 10^{-11}$ |
| **Previously reported genome-wide significant hits replicating in the follow-up** | | | | | | | | | | | | |
| 4 | HS3ST1 | rs4351014 | 11027619 | C | T | 0.684 | 0.94 [0.92–0.96] | $5.37 \times 10^{-10}$ | 0.93 [0.88–0.98] | $4.54 \times 10^{-3}$ | 0.94 [0.92–0.95] | $9.16 \times 10^{-12}$ |
| 16 | IL34 | rs4985556 | 70694000 | A | C | 0.111 | 1.08 [1.05–1.11] | $2.28 \times 10^{-8}$ | 1.09 [1.03–1.16] | $4.59 \times 10^{-3}$ | 1.08 [1.06–1.11] | $3.91 \times 10^{-10}$ |
| 16 | PLCG2 | rs12444183 | 81773209 | A | G | 0.407 | 0.95 [0.93–0.97] | $1.48 \times 10^{-8}$ | 0.92 [0.88–0.96] | $3.23 \times 10^{-5}$ | 0.95 [0.93–0.96] | $6.81 \times 10^{-12}$ |
| **Suggestive signals (not replicating)** | | | | | | | | | | | | |
| 14 | ELK2AP | rs7153315 | 106195719 | C | G | 0.750 | 0.94 [0.92–0.96] | $9.80 \times 10^{-8}$ | 1.16 [1.01–1.33] | 0.0412 | 0.94 [0.92–0.97] | $9.04 \times 10^{-7}$ |
| 15 | SPPL2A | rs76523702 | 51002342 | C | T | 0.802 | 1.06 [1.04–1.08] | $6.86 \times 10^{-8}$ | 1.02 [0.97–1.07] | 0.3501 | 1.05 [1.03–1.08] | $1.08 \times 10^{-7}$ |

Results obtained with a fixed effects inverse-variance-weighted meta-analysis on the discovery and follow-up stages. Freq A1 is from GR@ACE discovery dataset. P value for significance $<5 \times 10^{-8}$. Effect allele: A1.

PRS in the large GR@ACE dataset. The PRS was associated with a 1.27-fold (95% CI [1.23–1.32]) increased risk for every standard deviation increase in the PRS ($p = 7.3 \times 10^{-39}$) and with a gradual risk increase when we stratified the dataset into 2% percentiles of the PRS (Fig. 5a, Supplementary Data 6). Next, we stratified the dataset in *APOE* genotype risk groups. The PRS percentiles were associated with AD within the *APOE* genotype groups (Fig. 5b, Supplementary Data 7). Finally, we compared the risk extremes and found a 16.2-fold (95% CI [8.84–29.5], $p = 1.5 \times 10^{-19}$) increased risk for the highest-PRS group (*APOE ε4ε4*) compared with the lowest-PRS group (*APOE ε2ε2/ε2ε3*; Supplementary Data 8). When we compared the median AAO in AD patients in these extreme risk groups we found a 9-year difference in the median age ($p_{Wilcoxon} = 1.7 \times 10^{-6}$) (Fig. 5c). Lastly, we studied the effects on AAO of the PRS in the *APOE* genotype groups. The PRS differentiated AAO only within *APOE* ε4 carriers. In *APOE* ε4 heterozygotes the PRS determined a 4-year difference in median AAO and in *APOE* ε4 homozygotes ($p_{Wilcoxon} = 6.9 \times 10^{-5}$), where the PRS determined a median AAO difference of 5.5 years ($p_{Wilcoxon} = 4.6 \times 10^{-5}$). For the selection of high-risk individuals, it is important to note that we found no difference in the odds and AAO for AD for *APOE* ε4 heterozygotes with the highest PRS compared to *APOE* ε4 homozygotes with the lowest PRS. The Cox regression also showed an impact of *APOE* on AAO, mainly on *APOE* ε4ε4 (significant *APOE*-PRS interaction ($p = 0.021$), Fig. 5d, Supplementary Data 9).

## Discussion

This work adds on the ongoing global effort to identify genetic variants associated with AD (Fig. 3). In the present work, we reported on the largest GWAS for AD risk to date, comprising genetic information of 467,623 individuals of European ancestry. We identified six variants that were not previously associated with the risk of AD and constructed a robust PRS for AD demonstrating its potential value for selecting subjects at risk of AD, especially within *APOE* ε4 carriers. This PRS was based on European ancestries and may or may not generalize to other ancestries. Validation in other populations will be required. We also acknowledge that controls included in GR@ACE are younger than cases and some of the controls might still develop AD later in life. This fact does not invalidate the analysis although reported estimates must be considered conservative. The differences in risk and AAO determined by the PRS of AD are relevant for design clinical trials that over-represent *APOE* ε4 carriers, as *APOE* ε4 heterozygous with highest-PRS values have a similar risk and AAO to *APOE* ε4 homozygotes (Fig. 5b). These represents ~1% of our control population, which is the same percentage as all *APOE* ε4 homozygotes. A trial that aims to include *APOE* ε4 homozygotes, could consider widening the selection criteria and in this way hasten the enrollment process. Also, our PRS could aid at the interpretation of the results of clinical trials, as it determines a relevant proportion of the AAO, which could either mimic or obscure a treatment effect.

The most interesting finding from our GWAS is the discovery of a common protective (MAF (C-allele) = 0.483) intronic variant in the *APP* gene. Our results directly support *APP* production or processing as a causal pathway not only in familial AD but in common sporadic AD. The SNP is in a DNase hypersensitive area of 295 bp (chr21:27473781-27474075) possibly involved in the transcriptional regulation of the *APP* gene. rs2154481 is an eQTL for the *APP* mRNA and an antisense transcript of the *APP* gene named AP001439.2 in public eQTL databases[25] (Supplementary Fig. 4). Functional evidence supports a modified *APP* transcription[26] as an LD block of 13 SNPs within the *APP* locus

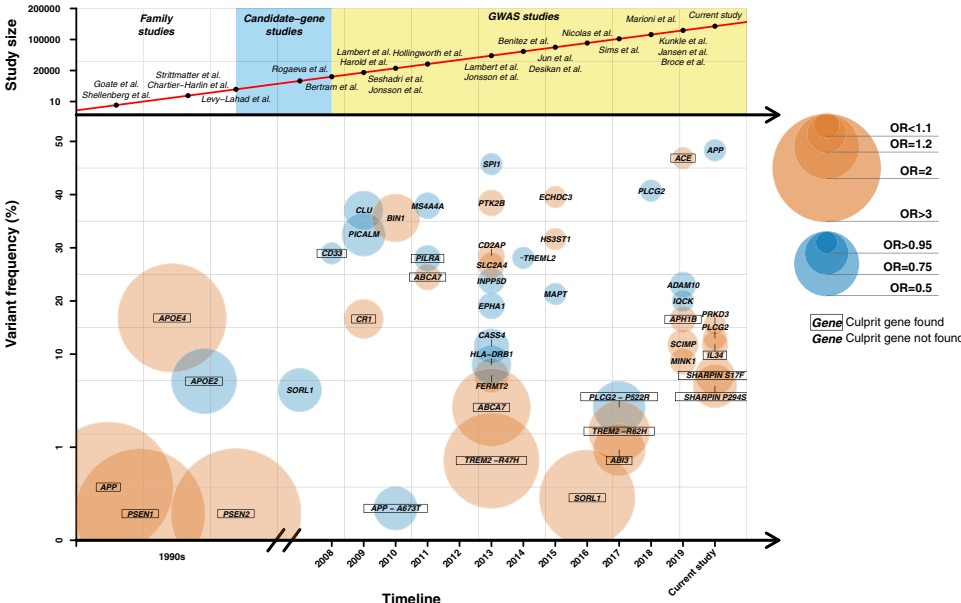

**Fig. 3 Genetic landscape for Alzheimer's disease.** This figure shows the history of genetic discoveries in AD research over the past 30 years. This figure was constructed to our best knowledge of literature, but is not a systematic review of literature. For common variants, we selected only signals firmly replicated in large meta-GWAS (Lambert et al.[3], Kunkle et al.[20], Jun et al.[43], Sims et al.[7], Jansen et al.[38] and present study). For rare variants, we only selected those variants widely replicated excluding those loci presenting conflicting results. Abbreviations and more information about the genes can be found in Supplementary Data 4. The risk alleles associated with AD were represented in orange and the protective alleles in blue. GWAS Genome-Wide Association Study, OR odds ratio.

(including rs2154481) increased the TFCP2 transcription factor avidity to its binding site and increased the enhancer activity of this specific intronic region[26]. Based on this evidence, we can postulate that a life-long slightly higher *APP* gene expression protects the brain from AD insults. Still, this seems counter-intuitive as duplications of the gene lead to early-onset AD[27]. A U-shaped effect, or hormesis effect of *APP* might help explain our observations and it might also fit the accelerated cognitive deterioration observed in AD patients treated with beta-secretase inhibitors[28,29] as these reduce beta-amyloid in their brain. An alternative hypothesis is that mechanisms underlying the variant are related to the overexpression of protective fragments of the APP protein[30]. Disentangling the molecular mechanism of our finding will help refine and steer the amyloid hypothesis.

Additionally, other three variants identified are altering protein sequence or affecting regulatory motifs. Two independent missense mutations in *SHARPIN* increased the AD risk. *SHARPIN* was previously proposed as an AD candidate gene[31,32], and functional analysis of a rare missense variant (NM_030974.3:p. Gly186Arg) resulted in the aberrant cellular localization of the variant protein and attenuated the activation of NF-κB, a central mediator of inflammatory and immune responses. Functional analysis of the two identified missense variants will show if the effect on immune reaction in AD is similar. The variant located in the *CHRNE* which encodes a subunit of the cholinergic receptor (AChR) is a strong modulator of *CHRNE* expression. The same allele that increases AD risk increases the expression in the brain and other tissues according to GTEx ($p = 2.1 \times 10^{-13}$) (Supplementary Fig. 5). The detection of a potential hypermorph allele linked to AD risk and affecting cholinergic function could reintroduce this neurotransmitter pathway into the search for preventative strategies. Further functional studies are needed to consolidate this hypothesis.

Altogether, we described six additional loci associated with sporadic AD. These signals reinforce that AD is a complex disease in which amyloid processing and immune response play key roles. We add to the growing body of evidence that the polygenic scores of all genetic loci to date, in combination with *APOE* genotypes, are robust tools that are associated with AD and its AAO. These properties make PRS promising in selecting individuals at risk to apply preventative therapeutic strategies.

## Methods
**Data.** Participants in this study were obtained from multiple sources, including raw data from case-control samples collected by GR@ACE/DEGESCO, summary statistics data from the case-control samples in the IGAP and the summary statistics of AD-by-proxy phenotype from the UK Biobank. An additional case-control samples from 16 independent cohorts (19,087 AD cases and 39,101 controls) was used for replication, largely collected and analyzed by the European Alzheimer's Disease Biobank (JPND-EADB) project. Full descriptions of the samples and their respective phenotyping and genotyping procedures are provided in the Supplementary Methods.

*GR@ACE*. The GR@ACE study[19] recruited AD patients from Fundació ACE, Institut Català de Neurociències Aplicades (Catalonia, Spain), and control individuals from three centers: Fundació ACE (Barcelona, Spain), Valme University Hospital (Seville, Spain), and the Spanish National DNA Bank–Carlos III (University of Salamanca, Spain) (http://www.bancoadn.org). Additional cases and controls were obtained from dementia cohorts included in the Dementia Genetics Spanish Consortium (DEGESCO)[33]. At all sites, AD diagnosis was established by a multidisciplinary working group—including neurologists, neuropsychologists, and social workers—according to the DSM-IV criteria for dementia and the National Institute on Aging and Alzheimer's Association's (NIA–AA) 2011 guidelines for diagnosing AD. In our study, we considered as AD cases any individuals with dementia diagnosed with probable or possible AD at any point in their clinical course. For further details on the contribution of the sites, see Supplementary Data 10. Written informed consent was obtained from all the participants. The ethics and scientific committees have approved this research protocol (Acta 25/2016, Ethics Committee H., Clinic I Provincial, Barcelona, Spain).

*Genotyping, quality control, and imputation.* DNA was extracted from peripheral blood according to standard procedures using the Chemagic system (Perkin Elmer). Samples reaching DNA concentrations of >10 ng/µl and presenting high integrity were included for genotyping. Cases and controls were randomized across sample plates to avoid batch effects.

Genotyping was conducted using the Axiom 815K Spanish biobank array (Thermo Fisher) at the Spanish National Center for Genotyping (CeGEN, Santiago de Compostela, Spain). The genotyping array not only is an adaptation of the Axiom biobank genotyping array but also contains rare population-specific

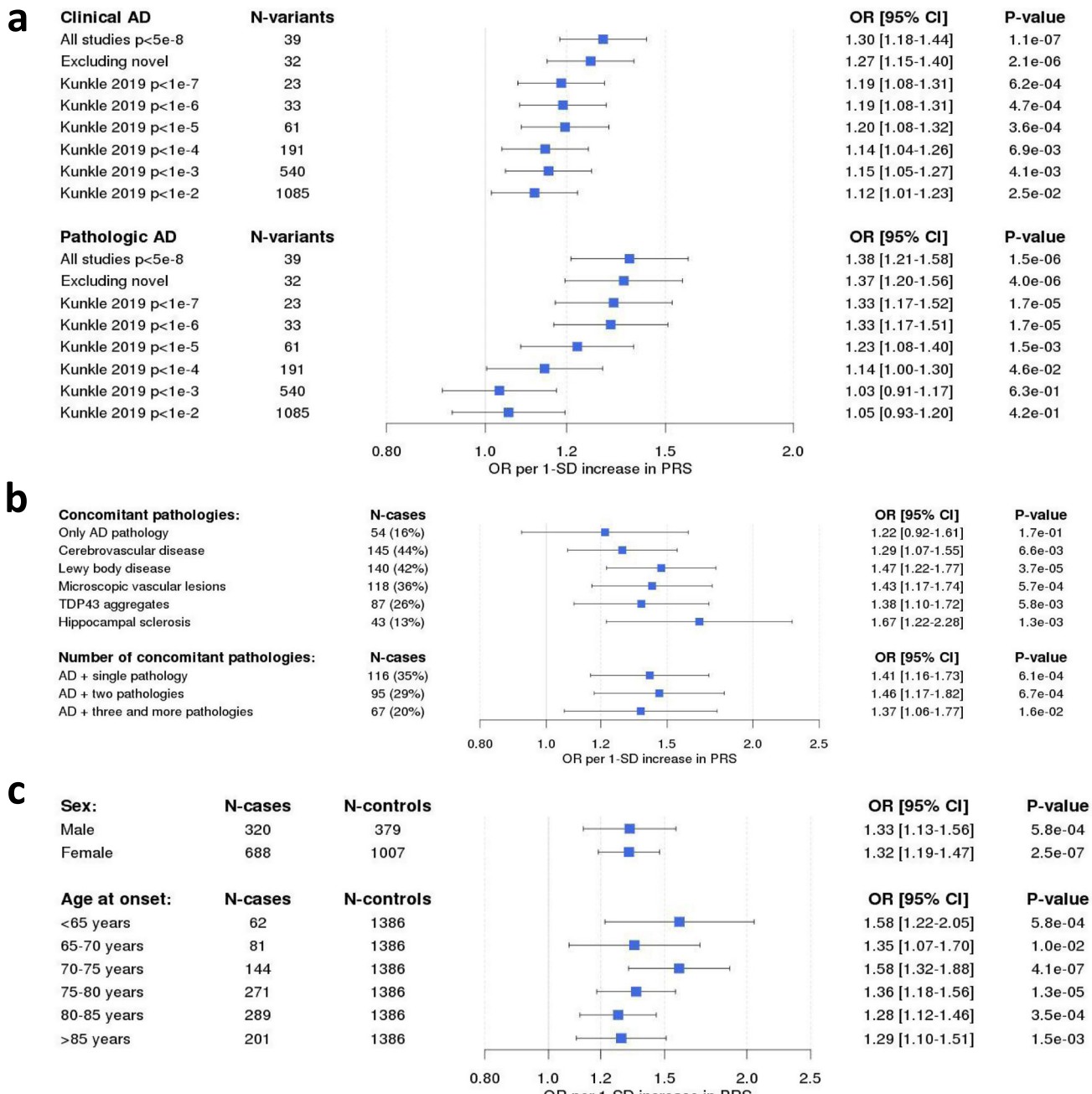

**Fig. 4 Polygenic risk scores for AD. a** The 39-SNP PRS association with clinical (OR = 1.30, 95% CI [1.18–1.44], $p = 1.1 \times 10^{-7}$) and pathologically confirmed AD cases (OR = 1.38, per 1-SD increase in the PRS, 95% CI [1.21–1.58], $p = 1.5 \times 10^{-6}$) from EADB-F.ACE/BBB dataset. **b** PRS association with AD in the presence of concomitant brain pathologies (besides AD). **c** PRS association with AD stratified by sex and AAO. A similar association of the PRS with AD was found in both sexes (OR$_{\text{males}}$ = 1.33, [1.13–1.56], $p = 5.8 \times 10^{-4}$ vs. OR$_{\text{females}}$ = 1.32, [1.19–1.47], $p = 2.5 \times 10^{-7}$). In (**a–c**) data are presented as Odds Ratio per 1-SD increase in PRS (95% CI). The generated PRS was validated using logistic regression adjusted by four principal components.

variations observed in the Spanish population. The DNA samples were genotyped according to the manufacturer's instructions (Axiom™ 2.0 Assay Manual Workflow). The Axiom 2.0 assay interrogates biallelic SNPs and simple indels in a single-assay workflow. Starting with 200 ng of genomic DNA, the samples were processed through a manual target preparation protocol, followed by automated processing of the array plates in the GeneTitan Multi-Channel (MC) instrument. Target preparation involved DNA amplification, fragmentation, purification, and resuspension of the target in a hybridization cocktail. The hyb-ready targets were then transferred to the GeneTitan MC instrument for automated, hands-free processing, including hybridization, staining, washing, and imaging. The CEL files were generated using the GeneTitan MC instrument. Quality control (QC) was performed for samples and plates using the Affymetrix power tool (APT) 1.15.0 software following the Axiom data analysis workflow. The sample quality

was determined based on the resolution of AT and GC channels in a group of non-polymorphic SNPs (resolution > 0.82). Samples with a call rate greater than 97% and plates with an average call rate above 98.5% were included for final SNP calling. The samples were jointly called. Markers passing all the QC tests were used in downstream analysis ($N_{\text{SNPs}} = 729,868$; 95.4%) using the SNPolisher R package (Thermo Fisher). To assess the sample genotyping concordance, we intentionally resampled 200 samples and determined a concordance rate of 99.5%.

We also conducted previously described standard QC prior to imputation[19]. In brief, individual QC includes genotype call rates >97%, sex checks, and no excess heterozygosity; we removed population outliers as well (European cluster of 1000 Genomes). We included variants with a call rate of >95%, with a minor allele frequency (MAF) of >0.01, in Hardy–Weinberg equilibrium ($p < 1 \times 10^{-4}$ in controls) and without differential missingness between cases and controls

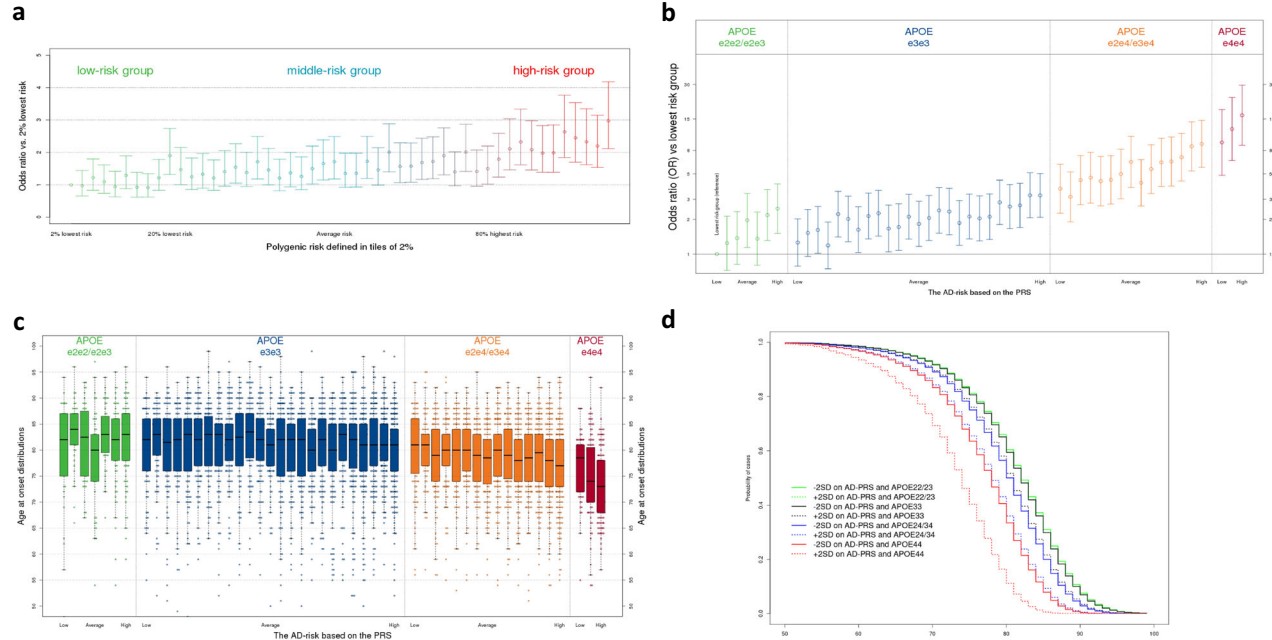

**Fig. 5 Polygenic Risk Scores *APOE* stratification for AD in *n* = 12,386 biologically independent samples from GR@ACE/DEGESCO. a** The AD risk of PRS groups compared to those with the 2% lowest risk. The 2% highest risk had a 3.0-fold (95% CI [2.12–4.18], $p = 3.2 \times 10^{-10}$) increased risk compared with those with the 2% lowest risk. No interaction was found between the PRS and *APOE* genotypes ($p$ value = 0.76). **b** The AD risk stratified by PRS and *APOE* risk groups compared to the lowest risk group (OR 95% CI). Association was found between highest and lowest-PRS percentiles within the *APOE* genotype groups: $\varepsilon2\varepsilon2/\varepsilon2\varepsilon3$ carriers (OR = 2.48 [1.51–4.08], $p = 3.4 \times 10^{-4}$), $\varepsilon3\varepsilon3$ carriers (OR = 2.67 [1.93–3.69], $p = 3.5 \times 10^{-9}$), $\varepsilon2\varepsilon4/\varepsilon3\varepsilon4$ carriers (OR = 2.47 [1.67–3.66], $p = 6.8 \times 10^{-6}$), and $\varepsilon4\varepsilon4$ carriers (OR = 2.02 [1.05–3.85], $p = 3.4 \times 10^{-2}$). Comparisons of the highest and lowest-PRS percentiles with respect to the *APOE* genotype groups: a difference was found between highest $\varepsilon2\varepsilon2/\varepsilon2\varepsilon3$ carriers vs. lowest $\varepsilon3\varepsilon3$ carriers (OR = 0.51 [0.34–0.75], $p = 7.8 \times 10^{-4}$), but not between highest $\varepsilon3\varepsilon3$ carriers vs. lowest $\varepsilon2\varepsilon4/\varepsilon3\varepsilon4$ carriers (OR = 1.17 [0.82–1.66], $p = 0.40$) and highest $\varepsilon2\varepsilon4/\varepsilon3\varepsilon4$ carriers vs. lowest $\varepsilon4\varepsilon4$ carriers (OR = 0.89 [0.52–1.53], $p = 0.68$). **c** The AAO of AD stratified by PRS and *APOE* risk groups. No difference in odds for AD was found between the PRS percentiles with AAO in *APOE* $\varepsilon2\varepsilon2/\varepsilon2\varepsilon3$ (lowest = 82 years, highest = 83 years, $p_{Wilcoxon} = 0.39$) and *APOE* $\varepsilon3\varepsilon3$ (lowest = 82 years, highest = 81 years, $p = 0.16$). However, a 4-year difference was found between *APOE* $\varepsilon4$ heterozygotes ($p_{Wilcoxon} = 6.9 \times 10^{-5}$, 81 years compared with 77 years) and 5.5 years difference ($p_{Wilcoxon} = 4.6 \times 10^{-5}$, 78.5 years compared with 73 years) in *APOE* $\varepsilon4$ homozygotes. Data are represented as boxplots as described in the manual of ggplot2 package in R. **a–c** Logistic regression models adjusted for four population ancestry components were used as statistical test. **d** Cox regression model on AAO. The determinants are the PRS and the *APOE* categories, a PRS*APOE interaction term and population substructure as covariates. The curve shows the probability a case in one of the eight groups has developed AD by a certain age (*x*-axis).

(Supplementary Data 11, Supplementary Fig. 1). Imputation was carried out using the Haplotype reference consortium[34] (HRC, full panel) and the 1000 Genomes reference panel[35] (for indels only) on the Michigan Imputation Server (https://imputationserver.sph.umich.edu). Rare variants (MAF < 0.001) and variants with low imputation quality ($R^2 < 0.30$) were excluded. Logistic regression models, adjusted for the first four ancestry principal components[19], were fitted using Plink (v2.00a). Population-based controls were used; therefore, age was not included as a covariate. Age and gender statistically behave like phenotype proxies (for AD status in this case). Therefore, adjusting for co-variation with age and gender could result in an over-adjustment of GWAS results. After QC steps, we included 6,331 AD cases and 6,055 control individuals and tested 14,542,816 genetic variants for association with AD.

*IGAP summary statistics.* The GWAS summary results from the IGAP were downloaded from the National Institute on Aging Genetics of Alzheimer's Disease Data Storage Site (NIAGADS, https://www.niagads.org/)[20]. Details on data generation and analyses by the IGAP have been previously described[20]. In brief, the IGAP is a large study based upon genome-wide association using individuals of European ancestry. Stage 1 of the IGAP comprises 21,982 AD cases and 41,944 cognitively normal controls from four consortia: the Alzheimer Disease Genetics Consortium (ADGC), the European Alzheimer's Disease Initiative (EADI), the Cohorts for Heart and Aging Research in Genomic Epidemiology (CHARGE) Consortium, and the Genetic and Environmental Risk in AD/Defining Genetic, Polygenic, and Environmental Risk for Alzheimer's Disease (GERAD/PERADES) Consortium. Summary statistics are available for 11,480,632 variants, both genotyped and imputed (1000 Genomes phase 1, v3). In Stage 2, 11,632 SNPs were genotyped in an independent set of 8362 AD cases and 10,483 controls.

*UK Biobank summary statistics.* UK Biobank data—including health, cognitive, and genetic data—was collected on over 500000 individuals aged 37–73 years from across Great Britain (England, Wales, and Scotland) at the study baseline

(2006–2010) (http://www.ukbiobank.ac.uk)[36]. Several groups have demonstrated the utility of self-report of parental history of AD for case ascertainment in GWAS (proxy–AD approach)[21,37,38]. For this study, we used the published summary statistics of Marioni et al.[21]. They included, after stringent QC, 314,278 unrelated individuals for whom AD information was available on at least one parent in the UK Biobank (https://datashare.is.ed.ac.uk/handle/10283/3364). In brief, the 27,696 participants whose mothers had dementia (maternal cases) were compared with the 260,980 participants whose mothers did not have dementia. Likewise, the 14,338 participants whose fathers had dementia (paternal cases) were compared with the 245,941 participants whose fathers did not have dementia[21]. The phenotype of the parents is independent, and therefore, the estimates could be meta-analyzed. After analysis, the effect estimates were made comparable to a case-control setting. Further information on the transformation of the effect sizes can be found elsewhere[21,39]. The data available comprises summary statistics of 7,794,553 SNPs imputed to the HRC reference panel (full panel).

**Meta-GWAS of AD.** After study-specific variant filtering and quality-control procedures, we performed a fixed effects inverse-variance–weighted meta-analysis[22] on the discovery and follow-up stages (Supplementary Data 1 and Supplementary Data 12). To determine the lead SNPs (those with the strongest association per genomic region), we performed clumping on SNPs with a GWS $p$ value ($p < 5 \times 10^{-8}$) (Plink v1.90, maximal linkage disequilibrium (LD) with $R^2 < 0.001$ and physical distance 250 Kb). In the *APOE* region, we only considered the *APOE* $\varepsilon4$ (rs429358) and *APOE* $\varepsilon2$ (rs7412) SNPs[40]. LD information was calculated using the GR@ACE imputed genotypes as a reference. Polygenicity and confounding biases, such as cryptic relatedness and population stratification, can yield an inflated distribution of test statistics in GWAS. To distinguish between inflation from a true polygenic signal and bias we quantified the contribution of each by examining the relationship between test statistics and linkage disequilibrium (LD) using the LD Score regression intercept (LDSC software[41]). Chromosomal regions associated with AD in previous studies were excluded from follow-up (Lambert

et al.[3], Kunkle et al.[42], and Jansen et al.[38]). We tested all variants with suggestive association ($p < 10^{-5}$) located in proximity (200 kb) of genomic regions selected for follow-up to allow for the potential refinement of the top associated variant.

Conditional analyses were performed in regions where multiple variants were associated with AD using logistic regression models, adjusting for the genetic variants in the region (Supplementary Data 13, 14).

Regional plots were generated with a mixture of homemade Python (v2.7) and R (v3.6.0) scripts. Briefly, given an input variant, we calculated the LD between the input variant and all the surrounding variants within a window of length defined by the user. The LD was calculated in the 1000 Genomes samples of European ancestry. We used gene positions from RefSeq (release 93); in the case of multiple gene models for a given gene, we reported the model with the largest number of exons. We used recombination rates from HapMap II and chromatin states from ENCODE/Broad (15 states were grouped to highlight the predicted functional elements). As a reference genome, we used GRCh37. Quantile–quantile plots, Manhattan plots, and the exploration of genomic inflation factors were performed using the R package qqman.

**Polygenic risk scores**. We calculated a weighted individual PRS based on the 39 genetic variants that showed GWS evidence of association with AD in the present study, excluding *APOE* to check the impact of PRS modulating *APOE* risk (Table 1 and Supplementary Data 3). The selected variants were directly genotyped or imputed with high quality (median imputation score $R^2 = 0.93$). The PRSs were generated by multiplying the genotype dosage of each risk allele for each variant by its respective weight and then summing across all variants. We weighted this by the effect size from previous IGAP studies [Kunkle et al.[42] (36 variants), Sims et al.[7] (2 variants), Jun et al.[43] (*MAPT* locus), Supplementary Data 5]. The generated PRS was validated using logistic regression adjusted by four principal components in a sample of 676 AD cases diagnosed based on clinical criteria and 332 pathologically confirmed AD cases from the European Alzheimer's Disease Biobank–Fundació ACE/Barcelona Brain Bank dataset (EADB–F.ACE/BBB, Supplementary Information). This dataset was not used in prior genetic studies. In this dataset, all pathologically confirmed cases were scored for the presence or absence of concomitant pathologies. In all analyses, we compared the AD patients to the same control dataset ($n = 1386$). We performed analyses to test the robustness of the PRS. We tested the effect of adding variants below the genome-wide significance threshold using a pruning and thresholding approach. For this, we used the summary statistics of the IGAP[42] study, and we selected independent variants using the clump_data() function from the TwoSampleMR package (v0.4.25). We used strict settings for clumping ($R^2 = 0.001$ and window = 1 MB) and increasing p value thresholds ($>1 \times 10^{-7}$, $>1 \times 10^{-6}$, $>1 \times 10^{-5}$, $>1 \times 10^{-4}$, $>1 \times 10^{-3}$, and $>1 \times 10^{-2}$). We tested the association of the results with clinically diagnosed and pathologically confirmed AD patients. To evaluate the effect of diagnostic certainty, we tested whether the PRS was different between the two patient groups. For the PRS with 39 GWS variants, we tested whether the PRS had sex-specific effects, whether it resulted in different age-of-onset groups of AD, and the effect of the PRS in the presence of concomitant brain pathologies.

*Risk stratification of the validated PRSs.* We searched for the groups at the highest risk of AD in the GR@ACE dataset (6331 AD cases and 6055 controls). We stratified the population into PRS percentiles, taking into account survival bias anticipated at old age[18]. To eliminate selection bias, we calculated the boundaries of the percentiles in the control participants aged 55 years and younger ($n = 3546$). Based on the boundaries from this population, the rest of the controls and all AD cases were then assigned into their appropriate percentiles. We first explored risk stratification using only the PRSs. For this, we split the PRSs into 50 groups (2 percentiles) and compared all groups with that which had the lowest PRS. Second, we explored risk stratification considering both the *APOE* genotypes and the PRSs. The *APOE* genotypes were pooled in the analyses as *APOE* ε2ε2/ε2ε3 ($n = 998$, split into 7 PRS groups), *APOE* ε3ε3 ($n = 7611$, split into 25 PRS groups), *APOE* ε2ε4/ε3ε4 ($n = 3399$, split into 15 PRS groups), and *APOE* ε4ε4 ($n = 382$, split into 3 PRS groups). We studied the effect of PRS across groups of individuals stratified by the *APOE* genotypes with the lowest-PRS group (*APOE* as the reference group using logistic regression models adjusted for four population ancestry components). Finally, we compared the median AAO using a Wilcoxon test.

We implemented a Cox regression model on AAO in the GR@ACE/DEGESCO dataset case-only adjusted for covariates as *APOE* group, the interaction between the PRS and *APOE* and four population ancestry components. All analyses were done in R (v3.4.2).

**Functional annotation**. We used Functional Mapping and Annotation of Genome-Wide Association Studies[23] (FUMA, v1.3.4c) to interpret SNP-trait associations (see Supplementary Methods and Supplementary Data 15–18). FUMA is an online platform that annotates GWAS findings and prioritizes the most likely causal SNPs and genes using information from 18 biological data repositories and tools. As input, we used the summary statistics of our meta-GWAS. Gene prioritization is based on a combination of positional mapping, expression quantitative trait loci (eQTL) mapping, and chromatin interaction mapping. Functional annotation was performed by applying a methodology similar to that described by Jansen et al.[38]. We referred to the original publication for details on the methods and repositories of FUMA[23].

**Reporting summary**. Further information on research design is available in the Nature Research Reporting Summary linked to this article.

## Data availability

The discovery summary statistics of this study are publicly available in Fundació ACE server [https://fundacioace-my.sharepoint.com/:u:/g/personal/iderojas_fundacioace_org/EaTwlPg9cRJHn7Kos4h39OUBaxajsjJHL_C110fC89bc8w?e=ZdcEUy].

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

## Acknowledgements

We would like to thank patients and controls who participated in this project. The present work has been performed as part of the doctoral program of I. de Rojas at the Universitat de Barcelona (Barcelona, Spain) supported by national grant from the Instituto de Salud Carlos III FI20/00215. The Genome Research @ Fundació ACE project (GR@ACE) is supported by Grifols SA, Fundació bancaria "La Caixa", Fundació ACE, and CIBERNED. A.R. and M.B. receive support from the European Union/EFPIA Innovative Medicines Initiative Joint undertaking ADAPTED and MOPEAD projects (grant numbers 115975 and 115985, respectively). M.B. and A.R. are also supported by national grants PI13/02434, PI16/01861, PI17/01474, PI19/01240 and PI19/01301. Acción Estratégica en Salud is integrated into the Spanish National R + D + I Plan and funded by ISCIII (Instituto de Salud Carlos III)—Subdirección General de Evaluación and the Fondo Europeo de Desarrollo Regional (FEDER—"Una manera de hacer Europa"). Some control samples and data from patients included in this study were provided in part by the National DNA Bank Carlos III (www.bancoadn.org, University of Salamanca, Spain) and Hospital Universitario Virgen de Valme (Sevilla, Spain); they were processed following standard operating procedures with the appropriate approval of the Ethical and Scientific Committee. Amsterdam dementia Cohort (ADC): Research of the Alzheimer center Amsterdam is part of the neurodegeneration research program of Amsterdam Neuroscience. The Alzheimer Center Amsterdam is supported by Stichting Alzheimer Nederland and Stichting VUmc fonds. The clinical database structure was developed with funding from Stichting Dioraphte. Genotyping of the Dutch case-control samples was performed in the context of EADB (European Alzheimer DNA biobank) funded by the JPco-fuND FP-829-029 (ZonMW project number 733051061). 100-Plus study: We are grateful for the collaborative efforts of all participating centenarians and their family members and/or relations. This work was supported by Stichting Alzheimer Nederland (WE09.2014-03), Stichting Dioraphte, horstingstuit foundation, Memorabel (ZonMW project number 733050814, 733050512) and Stichting VUmc Fonds. Genotyping of the 100-Plus Study was performed in the context of EADB (European Alzheimer DNA biobank) funded by the JPco-fuND FP-829-029 (ZonMW project number 733051061). Longitudinal Aging Study Amsterdam (LASA) is largely supported by a grant from the Netherlands Ministry of Health, Welfare and Sports, Directorate of Long-Term Care. The authors are grateful to all LASA participants, the fieldwork team and all researchers for their ongoing commitment to the study. This work was supported by a grant (European Alzheimer DNA BioBank, EADB) from the EU Joint Program—Neurodegenerative Disease Research (JPND) and also funded by Inserm, Institut Pasteur de Lille, the Lille Métropole Communauté Urbaine, the French government's LABEX DISTALZ program (development of innovative strategies for a transdisciplinary approach to AD). Genotyping of the German case-control samples was performed in the context of EADB (European Alzheimer DNA biobank) funded by the JPco-fuND (German Federal Ministry of Education and Research, BMBF: 01ED1619A). Full acknowledgments for the studies that contributed data can be found in the Supplementary Note. We thank the numerous participants, researchers, and staff from many studies who collected and contributed to the data. We thank the International Genomics of Alzheimer's Project (IGAP) for providing summary results data for these analyses. The investigators within IGAP contributed to the design and implementation of IGAP and/or provided data but did not participate in analysis or writing of this report. IGAP was made possible by the generous participation of the control subjects, the patients, and their families. The i–Select chips was funded by the French National Foundation on AD and related disorders. EADI was supported by the LABEX (laboratory of excellence program investment for the future) DISTALZ grant, Inserm, Institut Pasteur de Lille, Université de Lille 2 and the Lille University Hospital. GERAD was supported by the Medical Research Council (Grant n° 503480), Alzheimer's Research UK (Grant n° 503176), the Wellcome Trust (Grant n° 082604/2/07/Z) and German Federal Ministry of Education and Research (BMBF): Competence Network Dementia (CND) grant n° 01GI0102, 01GI0711, 01GI0420. CHARGE was partly supported by the NIA/NHLBI grants AG049505, AG058589, HL105756 and AGES contract N01–AG–12100, the Icelandic Heart Association, and the Erasmus Medical Center and Erasmus University. ADGC was supported by the NIH/NIA grants: U01 AG032984, U24 AG021886, U01 AG016976, and the Alzheimer's Association grant ADGC–10–196728. This research has been conducted using the UK Biobank public resource obtained through the University of Edinburg Data Share (https://datashare.is.ed.ac.uk/handle/10283/3364).

## Author contributions

A.Ru and S.v.d.L. designed and conceptualized the study, interpreted the data and drafted the paper. I.d.R. contributed to data acquisition, the analysis, interpreted the data, and co-wrote the paper. S.M.G. and N.T. contributed to the analysis and interpreted the data. S.J.v.d.L. and I.d.R. performed polygenic score analyses. L.C.C. and J.C. conducted the functional analysis of *APP*. H.Ho, W.v.d.F., S.J.v.d.L., and A.Ru supervised the study. All authors critically revised the paper for important intellectual content and approved the final paper. **GR@ACE/DEGESCO**: *Study design or conception*: M.Me, J.C., and A.Ru. *Data generation*: L.M., L.C.C., A.G.P., M.E.S., S.M.G., I.d.R., I.Q., and A.C. *Sample contribution*: M.Bo, J.M.G.A., M.Me, M.Ma, M.C., T.S., S.G.M., G.G.R., A.L.M., J.M., L. M.R., G.P.R., M.M.G., C.M.R., I.R.A., V.Al., C.L., E.R.R., P.S.J., D.Al., P.P., M.D.F., I.Al., J.P.T., A.C., L.Ta, A.M.M., M.J.B., A.F.G., I.Q., I.H., L.M., P.G.G., E.A.M., S.V., O.S.G., A. Bena, A.P.C., A.E., A.Sa, C.Ab, G.O., M.R.R., M.Al., N.R., S.G., A.O., A.Rab, A.Bel, F.Mo, M.Z., A.C.G., J.A.P., M.F.F., E.F.M., D.B.R., M.B.S.A., P.M., R.H.V., A.A.P., A.A., L.M.P., R.S.V., E.Ge, A.L., R.B., J.F., J.L.R., S.Men, M.Ba, I.d.R., and S.M.G. *Analysis*: S.M.G. and I.d.R. *Study supervision/management*: A.C., L.Ta, M.Bo, M.Me, J.C., and A.Ru. **IGAP**. *Critical revision*: A.C.N., B.W.K., L.A.F., J.L.H., L.S.W., M.A.P.V., R.May, M.A.I., J.C.B., A.L.D.S., C.L.S., E.B., M.F., Q.Y., X.J., R.S., C.H., K.M., S.Mea, V.E.P., A.Meg, P.A.H., R. Mar, P.A., G.Sc, J.Will, and S.Se. **EADB**: *Sample contribution*. N.T., I.E.J., N.St, K.A.M., C.D., G.N., G.C., G.Sp, K.S., M.I., M.K., R.F.S., J.C.L., A.Ra, D.G., J.S.V., D.R., E.Gr, H.Ha, I.G., J.K., L.Fa, L.Fr, A.M.H., J.V., L.H., G.H., N.Sca, M.H.K., M.Y., H.Ho, W.M.v.d.F, M. H.u.l., N.M.v.S., A.T.H., B.G.N., C.V.B., E.S., R.V., S.E., T.N., F.K., J.V.D., V.G., A.U., A. Benu, A.K.S., B.B., C.Mas, C.F., E.C., F.Ma, G.B., I.Ap, J.Q.T., L.Ki, L.K.l., L.P., L.Tr, L.B., M.L., M.Ar, R.G., S.F., F.J., J.D.S., O.G., T.G., M.J.H., T.P., K.Bu, M.E.si, S.R.H., E.Dur, A. Ru, I.H., S.M.G., I.d.R., Y.A.L..P, A.d.M., C.Cl., J.P., S.J.v.d.L., C.G., N.B., O.H., P.B., A. H., T.K., M.E.w., O.A.S., R.N.K., J.Wilf, P.F., P.R., P.Sc, P.Sa, N.Sch, D.W., E.R., G.R., H. S., I.R., A.Sc, A.Sp, A.Sq, C.Cha, C.Chi, C.P., A.P., B.A., B.N., C.M.F., D.S., E.Da, E.düz, E. F., F.T., F.P., F.S.G., G.Gi, G.Gra, G.P., H.B., J.H., J.L., M.C.D., M.T., M.T.H., M.Schm, M. W., M.S., O.Q., O.L., P.C., P.D., P.M., R.C., S.So, S.He, S.A., S.B., S.C., T.L., V.B., V.D., P. G.K., M.M.N., M.C.D.N., O.P., W.M., A.W.L., I.Ap, C.B.F., A.A.L.K., G.B., M.Sca, M.Sp, M.V., M.Hi, K.F., L.W., M.D., P.H., and A.Ra. *Analysis*: N.J.A., R.M.T., V.An, N.T., I.E.J., N.St, S.J.v.d.L., I.d.R., S.M.G., B.G.B., and C.B. *Studies supervision/management*: S.H.H., K.A.M., C.D., G.N., G.C., G.S., K.S., M.I., M.K, R.F.S., D.G., J.S.V., D.R., E.Gr, H.Ha, I.G., J.K., L.Fr, A.M.H., J.V., L.H., G.H., N.Sca, M.H.K., M.Y., H.Ho, W.M.v.d.F., M.Hui, N.M. v.S., N.J.A., J.D., M.Sche, A.K.S., C.G., N.B., O.H., P.B., A.H., T.K., J.Wilf, P.F., P.R., P.Sa, P.Sc, M.Hul, N.T., I.E.J., A.T.H., B.G.N., C.V.B., E.S., R.V., S.E., P.A., A.Ru, and J.C.L. **PGC-ALZ**: *Sample contribution*. I.E.J., A.Ro, I.Sa, D.Aa, G.Se, S.B.S., S.D., D.P., S.H., I.K. K., N.L.P., C.A.R., and O.A.A. *Analysis*: S.Ha, I.K.K., and O.A.A. *Study supervision/management*: O.A.A., C.M.v.D., and D.P. **AD and GBCS**: *Sample contribution*. H.Z., S.K., I.S., and K.B. *Analysis*: N.M.S. and A.Z. *Study supervision/management*: I.S.k., A.Z., and K.B.l. **NxC**: *Sample contribution*. A.C.A., M.T.M., M.S.R., and C.An.

## Competing interests

The authors declare no competing interests.

## Additional information

Itziar de Rojas[1,2,306], Sonia Moreno-Grau[1,2,306], Niccolo Tesi[3,4,5,306], Benjamin Grenier-Boley[6,306], Victor Andrade[7,8,306], Iris E. Jansen[3,9,306], Nancy L. Pedersen[10], Najada Stringa[11], Anna Zettergren[12], Isabel Hernández[1,2], Laura Montrreal[1], Carmen Antúnez[13], Anna Antonell[14], Rick M. Tankard[15], Joshua C. Bis[16], Rebecca Sims[17], Céline Bellenguez[6], Inés Quintela[18], Antonio González-Perez[19], Miguel Calero[2,20,21], Emilio Franco-Macías[22], Juan Macías[23], Rafael Blesa[2,24], Laura Cervera-Carles[2,24], Manuel Menéndez-González[25,26,27], Ana Frank-García[2,28,29,30], Jose Luís Royo[31], Fermin Moreno[2,32,33], Raquel Huerto Vilas[34,35], Miquel Baquero[36], Mónica Diez-Fairen[37,38], Carmen Lage[2,39], Sebastián García-Madrona[40], Pablo García-González[1], Emilio Alarcón-Martín[1,31], Sergi Valero[1,2], Oscar Sotolongo-Grau[1], Abbe Ullgren[41,42], Adam C. Naj[43,44], Afina W. Lemstra[3], Alba Benaque[1], Alba Pérez-Cordón[1], Alberto Benussi[45], Alberto Rábano[2,21,46], Alessandro Padovani[45], Alessio Squassina[47], Alexandre de Mendonça[48], Alfonso Arias Pastor[34,35], Almar A. L. Kok[11,49], Alun Meggy[50], Ana Belén Pastor[21,46], Ana Espinosa[1,2], Anaïs Corma-Gómez[23], Angel Martín Montes[2,29,51], Ángela Sanabria[1,2], Anita L. DeStefano[52,53], Anja Schneider[8,54], Annakaisa Haapasalo[55], Anne Kinhult Ståhlbom[41,42], Anne Tybjærg-Hansen[56,57], Annette M. Hartmann[58], Annika Spottke[54,59], Arturo Corbatón-Anchuelo[60,61], Arvid Rongve[62,63], Barbara Borroni[45], Beatrice Arosio[64,65], Benedetta Nacmias[66,67], Børge G. Nordestgaard[57,68], Brian W. Kunkle[69,70], Camille Charbonnier[71], Carla Abdelnour[1,2], Carlo Masullo[72], Carmen Martínez Rodríguez[26,73], Carmen Muñoz-Fernandez[74], Carole Dufouil[75,76], Caroline Graff[41,42], Catarina B. Ferreira[77], Caterina Chillotti[78], Chandra A. Reynolds[79], Chiara Fenoglio[80], Christine Van Broeckhoven[81,82,83], Christopher Clark[84], Claudia Pisanu[47], Claudia L. Satizabal[52,85,86], Clive Holmes[87], Dolores Buiza-Rueda[2,88], Dag Aarsland[89,90], Dan Rujescu[58], Daniel Alcolea[2,24], Daniela Galimberti[80,91], David Wallon[92], Davide Seripa[93], Edna Grünblatt[94,95,96], Efthimios Dardiotis[97], Emrah Düzel[98,99], Elio Scarpini[80,91], Elisa Conti[100], Elisa Rubino[101], Ellen Gelpi[102,103], Eloy Rodriguez-Rodriguez[2,39], Emmanuelle Duron[104,105,106], Eric Boerwinkle[107,108], Evelyn Ferri[65], Fabrizio Tagliavini[109], Fahri Küçükali[81,82,83], Florence Pasquier[110,111], Florentino Sanchez-Garcia[112], Francesca Mangialasche[113], Frank Jessen[54,114,115], Gaël Nicolas[73], Geir Selbæk[116,117,118], Gemma Ortega[1,2], Geneviève Chêne[75,76], Georgios Hadjigeorgiou[119], Giacomina Rossi[109], Gianfranco Spalletta[120,121], Giorgio Giaccone[109], Giulia Grande[122], Giuliano Binetti[123,124], Goran Papenberg[122], Harald Hampel[125], Henri Bailly[106,126], Henrik Zetterberg[127,128,129,130], Hilkka Soininen[131,132], Ida K. Karlsson[10,133], Ignacio Alvarez[37,38], Ildebrando Appollonio[100,134], Ina Giegling[58], Ingmar Skoog[12], Ingvild Saltvedt[135,136], Innocenzo Rainero[137], Irene Rosas Allende[26,138], Jakub Hort[139,140], Janine Diehl-Schmid[141], Jasper Van Dongen[81,82], Jean-Sebastien Vidal[106,126], Jenni Lehtisalo[131,142], Jens Wiltfang[143,144,145],

Jesper Qvist Thomassen[56], Johannes Kornhuber [146], Jonathan L. Haines [147,148], Jonathan Vogelgsang [143,149], Juan A. Pineda [23], Juan Fortea[2,24], Julius Popp[150,151,152], Jürgen Deckert[153], Katharina Buerger[154,155], Kevin Morgan [156], Klaus Fließbach[8], Kristel Sleegers [81,82,83], Laura Molina-Porcel[14,102], Lena Kilander[157], Leonie Weinhold[158], Lindsay A. Farrer [159], Li-San Wang[44], Luca Kleineidam[7,8], Lucia Farotti[160], Lucilla Parnetti[160], Lucio Tremolizzo[100,134], Lucrezia Hausner[161], Luisa Benussi[124], Lutz Froelich[161], M. Arfan Ikram [162], M. Candida Deniz-Naranjo[112], Magda Tsolaki [163], Maitée Rosende-Roca[1,2], Malin Löwenmark[157], Marc Hulsman[3,4], Marco Spallazzi[164], Margaret A. Pericak-Vance [70], Margaret Esiri[165], María Bernal Sánchez-Arjona[22], Maria Carolina Dalmasso[7], María Teresa Martínez-Larrad[60,61], Marina Arcaro[91], Markus M. Nöthen[166], Marta Fernández-Fuertes[23], Martin Dichgans [154,155,167], Martin Ingelsson[157], Martin J. Herrmann[153], Martin Scherer[168], Martin Vyhnalek[139,140], Mary H. Kosmidis[169], Mary Yannakoulia[170], Matthias Schmid[54,158], Michael Ewers [154,155], Michael T. Heneka[8,54], Michael Wagner [8,54], Michela Scamosci[171], Miia Kivipelto[113,172,173,174], Mikko Hiltunen[175], Miren Zulaica[2,33], Montserrat Alegret[1,2], Myriam Fornage[176], Natalia Roberto[1], Natasja M. van Schoor[11], Nazib M. Seidu[12], Nerisa Banaj[120], Nicola J. Armstrong[15], Nikolaos Scarmeas[177,178], Norbert Scherbaum[179], Oliver Goldhardt[141], Oliver Hanon[106,126], Oliver Peters[180,181], Olivia Anna Skrobot[182], Olivier Quenez [71], Ondrej Lerch[139,140], Paola Bossù [183], Paolo Caffarra[184], Paolo Dionigi Rossi[65], Paraskevi Sakka[185], Patrizia Mecocci[171], Per Hoffmann[166,186], Peter A. Holmans [17], Peter Fischer[187], Peter Riederer[188], Qiong Yang [53], Rachel Marshall[17], Rajesh N. Kalaria[189,190], Richard Mayeux[191,192,193], Rik Vandenberghe[194,195], Roberta Cecchetti[171], Roberta Ghidoni[124], Ruth Frikke-Schmidt[56,57], Sandro Sorbi[66,67], Sara Hägg [10], Sebastiaan Engelborghs[196,197,198,199], Seppo Helisalmi[200], Sigrid Botne Sando[201,202], Silke Kern[12], Silvana Archetti[203], Silvia Boschi[137], Silvia Fostinelli[124], Silvia Gil[1], Silvia Mendoza[204], Simon Mead[205], Simona Ciccone[65], Srdjan Djurovic [206,207], Stefanie Heilmann-Heimbach[166], Steffi Riedel-Heller[208], Teemu Kuulasmaa[175], Teodoro del Ser[209], Thibaud Lebouvier[110,111], Thomas Polak[153], Tiia Ngandu[113,142], Timo Grimmer[141], Valentina Bessi [66,210], Valentina Escott-Price [17,211], Vilmantas Giedraitis [157], Vincent Deramecourt[110,111], Wolfgang Maier[8,54], Xueqiu Jian [85], Yolande A. L. Pijnenburg[3], EADB contributors*, The GR@ACE study group*, DEGESCO consortium*, IGAP (ADGC, CHARGE, EADI, GERAD)*, PGC-ALZ consortia*, Patrick Gavin Kehoe [182], Guillermo Garcia-Ribas[40], Pascual Sánchez-Juan [2,39], Pau Pastor [37,38], Jordi Pérez-Tur [2,212,213], Gerard Piñol-Ripoll[34,35], Adolfo Lopez de Munain [2,32,33,214], Jose María García-Alberca [2,204], María J. Bullido [2,30,215,216], Victoria Álvarez[26,138], Alberto Lleó [2,24], Luis M. Real [23,217], Pablo Mir[2,88], Miguel Medina [2,21], Philip Scheltens[3], Henne Holstege[3,4], Marta Marquié[1,2], María Eugenia Sáez [19], Ángel Carracedo[18,218], Philippe Amouyel [6], Gerard D. Schellenberg[44], Julie Williams [17,50], Sudha Seshadri [52,85,219], Cornelia M. van Duijn [162,220], Karen A. Mather [221,222], Raquel Sánchez-Valle[14], Manuel Serrano-Ríos[60,61], Adelina Orellana[1,2], Lluís Tárraga[1,2], Kaj Blennow [127,128], Martijn Huisman[11,223], Ole A. Andreassen [224,225], Danielle Posthuma[9], Jordi Clarimón[2,24,307], Mercè Boada [1,2,307], Wiesje M. van der Flier [3,307], Alfredo Ramirez [7,8,54,226], Jean-Charles Lambert [6,307], Sven J. van der Lee [3,4,307]✉ & Agustín Ruiz [1,2,307]✉

[1]Research Center and Memory clinic Fundació ACE, Institut Català de Neurociències Aplicades, Universitat Internacional de Catalunya, Barcelona, Spain. [2]CIBERNED, Network Center for Biomedical Research in Neurodegenerative Diseases, National Institute of Health Carlos III, Madrid, Spain. [3]Alzheimer Center Amsterdam, Department of Neurology, Amsterdam Neuroscience, Vrije Universiteit Amsterdam, Amsterdam UMC, Amsterdam, The Netherlands. [4]Section Genomics of Neurodegenerative Diseases and Aging, Department of Clinical Genetics, Vrije Universiteit Amsterdam, Amsterdam UMC, Amsterdam, The Netherlands. [5]Delft Bioinformatics Lab, Delft Univeristy of Technology, Delft, The Netherlands. [6]Univ. Lille, Inserm, Institut Pasteur de Lille, CHU Lille, U1167-Labex DISTALZ-RID-AGE-Risk Factors and Molecular Determinants of Aging-Related Diseases, Lille, France. [7]Division of Neurogenetics and Molecular Psychiatry, Department of Psychiatry and Psychotherapy, University of Cologne, Medical Faculty, Cologne, Germany. [8]Department of Neurodegenerative diseases and Geriatric Psychiatry, University Clinic Bonn, Bonn, Germany. [9]Department of Complex Trait Genetics, Center for Neurogenomics and Cognitive Research, Amsterdam Neuroscience, VU University,

Amsterdam, The Netherlands. [10]Department of Medical Epidemiology and Biostatistics, Karolinska Institutet, Stockholm, Sweden. [11]Amsterdam UMC-Vrije Universiteit Amsterdam, Department of Epidemiology and Data Science, Amsterdam Public Health Research Institute, Amsterdam, The Netherlands. [12]Neuropsychiatric Epidemiology Unit, Department of Psychiatry and Neurochemistry, Institute of Neuroscience and Physiology, Sahlgrenska Academy, Centre for Ageing and Health (AgeCap), University of Gothenburg, Gothenburg, Sweden. [13]Unidad de Demencias, Hospital Clínico Universitario Virgen de la Arrixaca, Murcia, Spain. [14]Alzheimer's disease and other cognitive disorders unit. Service of Neurology, Hospital Clínic of Barcelona. Institut d'Investigacions Biomèdiques August Pi i Sunyer, University of Barcelona, Barcelona, Spain. [15]Mathematics and Statistics, Murdoch University, Perth, WA, Australia. [16]Cardiovascular Health Research Unit, Department of Medicine, University of Washington, Seattle, WA, USA. [17]Division of Psychological Medicine and Clinial Neurosciences, MRC Centre for Neuropsychiatric Genetics and Genomics, Cardiff University, Cardiff, UK. [18]Grupo de Medicina Xenómica, Centro Nacional de Genotipado (CEGEN-PRB3-ISCIII), Universidade de Santiago de Compostela, Santiago de Compostela, Spain. [19]CAEBI, Centro Andaluz de Estudios Bioinformáticos, Sevilla, Spain. [20]UFIEC, Instituto de Salud Carlos III, Madrid, Spain. [21]CIEN Foundation/Queen Sofia Foundation Alzheimer Center, Madrid, Spain. [22]Unidad de Demencias, Servicio de Neurología y Neurofisiología, Instituto de Biomedicina de Sevilla (IBiS), Hospital Universitario Virgen del Rocío/CSIC/Universidad de Sevilla, Sevilla, Spain. [23]Unidad Clínica de Enfermedades Infecciosas y Microbiología, Hospital Universitario de Valme, Sevilla, Spain. [24]Department of Neurology, II B Sant Pau, Hospital de la Santa Creu i Sant Pau, Universitat Autònoma de Barcelona, Barcelona, Spain. [25]Servicio de Neurología, Hospital Universitario Central de Asturias, Oviedo, Spain. [26]Instituto de Investigación Sanitaria del Principado de Asturias (ISPA), Oviedo, Spain. [27]Departamento de Medicina, Universidad de Oviedo, Oviedo, Spain. [28]Department of Neurology, La Paz University Hospital, Instituto de Investigación Sanitaria del Hospital Universitario La Paz, IdiPAZ, Madrid, Spain. [29]Hospital La Paz Institute for Health Research, IdiPAZ, Madrid, Spain. [30]Universidad Autónoma de Madrid, Madrid, Spain. [31]Departamento de Especialidades Quirúrgicas, Bioquímicas e Inmunología, School of Medicine, University of Málaga, Málaga, Spain. [32]Department of Neurology, Hospital Universitario Donostia, San Sebastian, Spain. [33]Neurosciences Area, Instituto Biodonostia, San Sebastian, Spain. [34]Unitat Trastorns Cognitius, Hospital Universitari Santa Maria de Lleida, Lleida, Spain. [35]Institut de Recerca Biomedica de Lleida (IRBLLeida), Lleida, Spain. [36]Servei de Neurologia, Hospital Universitari i Politècnic La Fe, Valencia, Spain. [37]Fundació Docència i Recerca MútuaTerrassa, Terrassa, Barcelona, Spain. [38]Memory Disorders Unit, Department of Neurology, Hospital Universitari Mutua de Terrassa, Terrassa, Barcelona, Spain. [39]Neurology Service, Marqués de Valdecilla University Hospital (University of Cantabria and IDIVAL), Santander, Spain. [40]Hospital Universitario Ramon y Cajal, IRYCIS, Madrid, Spain. [41]Karolinska Institutet, Center for Alzheimer Research, Department NVS, Division of Neurogeriatrics, Stockholm, Sweden. [42]Unit for Hereditary Dementias, Theme Aging, Karolinska University Hospital-Solna, Stockholm, Sweden. [43]Department of Biostatistics, Epidemiology and Informatics, University of Pennsylvania Perelman School of Medicine, Philadelphia, PA, USA. [44]Penn Neurodegeneration Genomics Center, Department of Pathology and Laboratory Medicine, University of Pennsylvania Perelman School of Medicine, Philadelphia, PA, USA. [45]Centre for Neurodegenerative Disorders, Department of Clinical and Experimental Sciences, University of Brescia, Brescia, Italy. [46]BT-CIEN, Madrid, Spain. [47]Department of Biomedical Sciences, Section of Neuroscience and Clinical Pharmacology, University of Cagliari, Cagliari, Italy. [48]Faculty of Medicine, University of Lisbon, Lisbon, Portugal. [49]Amsterdam UMC, Vrije Universiteit Amsterdam, Department of Psychiatry, Amsterdam Public Health Research Institute, Amsterdam, The Netherlands. [50]UK Dementia Research Institute at Cardiff, Cardiff University, Cardiff, UK. [51]Department of Neurology, La Paz University Hospital, Madrid, Spain. [52]Department of Neurology, Boston University School of Medicine, Boston, MA, USA. [53]Department of Biostatistics, Boston University School of Public Health, Boston, MA, USA. [54]German Center for Neurodegenerative Diseases (DZNE), Bonn, Germany. [55]A.I Virtanen Institute for Molecular Sciences, University of Eastern Finland, Kuopio, Finland. [56]Department of Clinical Biochemistry, Rigshospitalet, Copenhagen, Denmark. [57]Department of Clinical Medicine, Faculty of Health and Medical Sciences, University of Copenhagen, Copenhagen, Denmark. [58]Martin-Luther-University Halle-Wittenberg, University Clinic and Outpatient Clinic for Psychiatry, Psychotherapy and Psychosomatics, Halle (Saale), Germany. [59]Department of Neurology, University of Bonn, Bonn, Germany. [60]Instituto de Investigación Sanitaria, Hospital Clínico San Carlos (IdISSC), Madrid, Spain. [61]Spanish Biomedical Research Centre in Diabetes and Associated Metabolic Disorders (CIBERDEM), Madrid, Spain. [62]Haugesund Hospital, Helse Fonna, Department of Research and Innovation, Haugesund, Norway. [63]University of Bergen, Institute of Clinical Medicine (K1), Bergen, Norway. [64]Department of Clinical Sciences and Community Health, University of Milan, Milan, Italy. [65]Geriatric Unit, Fondazione Cà Granda, IRCCS Ospedale Maggiore Policlinico, Milan, Italy. [66]Department of Neuroscience, Psychology, Drug Research and Child Health University of Florence, Florence, Italy. [67]IRCCS Fondazione Don Carlo Gnocchi, Florence, Italy. [68]Department of Clinical Biochemistry, Herlev Gentofte Hospital, Herlev, Denmark. [69]Dr. John T. Macdonald Foundation Department of Human Genetics, University of Miami Miller School of Medicine, Miami, FL, USA. [70]John P. Hussman Institute for Human Genomics, University of Miami Miller School of Medicine, Miami, FL, USA. [71]Normandie Univ, UNIROUEN, Inserm U1245, CHU Rouen, Department of Genetics and CNR-MAJ, FHU G4 Génomique, F-76000 Rouen, France. [72]Institute of Neurology, Catholic University of the Sacred Heart, School of Medicine, Milan, Italy. [73]Hospital de Cabueñes, Gijón, Spain. [74]Servicio de Neurología, Hospital Universitario de Gran Canaria Dr.Negrín, Las Palmas, Spain. [75]Inserm, Bordeaux Population Health Research Center, UMR 1219, Univ. Bordeaux, ISPED, CIC 1401-EC, Univ Bordeaux, Bordeaux, France. [76]CHU de Bordeaux, Pole de Santé Publique, Bordeaux, France. [77]Instituto de Medicina Molecular João lobo Antunes, Faculdade de Medicina, Universidade de Lisboa, Lisboa, Portugal. [78]Unit of Clinical Pharmacology, University Hospital of Cagliari, Cagliari, Italy. [79]Department of Psychology, University of California—Riverside, Riverside, CA, USA. [80]University of Milan, Dino Ferrari Center, Milan, Italy. [81]VIB Center for Molecular Neurology, Antwerp, Belgium. [82]Laboratory of Neurogenetics, Institute Born-Bunge, Antwerp, Belgium. [83]Department of Biomedical Sciences, University of Antwerp., Antwerp, Belgium. [84]Insititute for Regenerative Medicine, University of Zürich, Zürich, Switzerland. [85]Glenn Biggs Institute for Alzheimer's and Neurodegenerative Diseases, San Antonio, TX, USA. [86]Department of Population Health Sciences, UT Health San Antonio, San Antonio, TX, USA. [87]Division of Clinical Neurosciences, School of Medicine, University of Southampton, Southampton, UK. [88]Unidad de Trastornos del Movimiento, Servicio de Neurología y Neurofisiología, Instituto de Biomedicina de Sevilla (IBiS), Hospital Universitario Virgen del Rocío/CSIC/Universidad de Sevilla, Sevilla, Spain. [89]Department of Old Age Psychiatry, Institute of Psychiatry, Psychology & Neuroscience, King's College London, London, UK. [90]Centre of Age-Related Medicine, Stavanger University Hospital, Stavanger, Norway. [91]Fondazione IRCCS Ca' Granda, Ospedale Policlinico, Milan, Italy. [92]Normandie Univ, UNIROUEN, Inserm U1245, CHU Rouen, Department of Neurology and CNR-MAJ, FHU G4 Génomique, F-76000 Rouen, France. [93]Complex Structure of Geriatrics, Department of Medical Sciences Fondazione IRCCS Casa Sollievo della Sofferenza, San Giovanni Rotondo (FG), Italy. [94]Department of Child and Adolescent Psychiatry and Psychotherapy, Psychiatric University Hospital Zurich (PUK), University of Zurich, Zurich, Switzerland. [95]Neuroscience Center Zurich, University of Zurich and ETH Zurich, Zurich, Switzerland. [96]Zurich Center for Integrative Human Physiology, University of Zurich, Zurich, Switzerland. [97]School of Medicine, University of Thessaly, Larissa, Greece. [98]German Center for Neurodegenerative Diseases (DZNE), Magdeburg, Germany. [99]Institute of Cognitive Neurology and Dementia Research (IKND), Otto-von-Guericke University, Magdeburg, Germany. [100]School of Medicine and Surgery, University of Milano-Bicocca and Milan Center for Neuroscience, Milan, Italy. [101]Department of Neuroscience and Mental Health, AOU Città della Salute e della Scienza di Torino, Torino, Italy. [102]Neurological Tissue Bank of the Biobanc-Hospital Clinic-IDIBAPS, Institut d'Investigacions Biomèdiques August Pi i Sunyer, Barcelona, Spain. [103]Division of Neuropathology and Neurochemistry, Department of Neurology, Medical University of Vienna, Vienna, Austria. [104]APHP, Hôpital

Brousse, equipe INSERM 1178, MOODS, Villejuif, France. [105]Université Paris-Saclay, UVSQ, Inserm, CESP, Team MOODS, Le Kremlin-Bicêtre, Paris, France. [106]APHP, Hôpital Broca, Paris, France. [107]School of Public Health, Human Genetics Center, University of Texas Health Science Center at Houston, Houston, TX, USA. [108]Human Genome Sequencing Center, Baylor College of Medicine, Houston, TX, USA. [109]Fondazione IRCCS Istituto Neurologico Carlo Besta, Milan, Italy. [110]Inserm U1172, CHU, DISTAlz, LiCEND, Univ Lille, Lille, France. [111]CHU CNR-MAJ, Lille, France. [112]Servicio de Inmunología, Hospital Universitario de Gran Canaria Dr. Negrín, Las Palmas de Gran Canaria, Spain. [113]Division of Clinical Geriatrics, Center for Alzheimer Research, Department of Neurobiology, Care Sciences and Society (NVS), Karolinska Institutet, Stockholm, Sweden. [114]Department of Psychiatry and Psychotherapy, University of Cologne, Medical Faculty, Cologne, Germany. [115]Excellence Cluster on Cellular Stress Responses in Aging-Associated Diseases (CECAD), University of Cologne, Cologne, Germany. [116]Norwegian National Advisory Unit on Ageing and Health, Vestfold Hospital Trust, Tønsberg, Norway. [117]Department of Geriatric Medicine, Oslo University Hospital, Oslo, Norway. [118]Institute of Clinical Medicine, University of Oslo, Oslo, Norway. [119]Department of Neurology, Medical School, University of Cyprus, Nicosia, Cyprus. [120]Laboratory of Neuropsychiatry, IRCCS Santa Lucia Foundation, Rome, Italy. [121]Beth K. and Stuart C. Yudofsky Division of Neuropsychiatry, Department of Psychiatry and Behavioral Sciences, Baylor College of Medicine, Houston, TX, USA. [122]Aging Research Center, Department of Neurobiology, Care Sciences and Society, Karolinska Institutet and Stockholm University, Stockholm, Sweden. [123]MAC—Memory Clinic, IRCCS Istituto Centro San Giovanni di Dio Fatebenefratelli, Brescia, Italy. [124]Molecular Markers Laboratory, IRCCS Istituto Centro San Giovanni di Dio Fatebenefratelli, Brescia, Italy. [125]Sorbonne University, GRC n° 21, Alzheimer Precision Medicine (APM), AP-HP, Pitié-Salpêtrière Hospital, Paris, France. [126]EA 4468, Sorbonne Paris Cité, Université Paris Descartes, Paris, France. [127]Clinical Neurochemistry Laboratory, Sahlgrenska University Hospital, Mölndal, Sweden. [128]Department of Psychiatry and Neurochemistry, Institute of Neuroscience and Physiology, Sahlgrenska Academy at the University of Gothenburg, Gothenburg, Sweden. [129]Department of Neurodegenerative Disease, UCL Institute of Neurology, London, UK. [130]UK Dementia Research Institute at UCL, London, UK. [131]Institute of Clinical Medicine Neurology,  University of Eastern Finland, Kuopio, Finland. [132]Neurocenter, neurology, Kuopio University Hospital, Kuopio, Finland. [133]Institute for Gerontology and Aging Research Network—Jönköping (ARN-J), School of Health and Welfare, Jönköping University, Jönköping, Sweden. [134]Neurology Unit, 'San Gerardo' hospital, Monza, Italy. [135]Department of Geriatrics, Clinic of Medicine, St Olavs Hospital, University Hospital of Trondheim, Trondheim, Norway. [136]Department of Neuromedicine and Movement Science, Norwegian University of Science and Technhology (NTNU), Trondheim, Norway. [137]Department of Neuroscience "Rita Levi Montalcini", University of Torino, Torino, Italy. [138]Laboratorio de Genética, Hospital Universitario Central de Asturias, Oviedo, Spain. [139]Memory Clinic, Department of Neurology, 2nd Faculty of Medicine and Motol University Hospital, Charles University, Prague, Czech Republic. [140]International Clinical Research Center, St. Anne's University Hospital Brno, Brno, Czech Republic. [141]Department of Psychiatry and Psychotherapy, School of Medicine Klinikum rechts der Isar,  Technical University of Munich, Munich, Germany. [142]Population Health Unit, Finnish Institute for Health and Welfare, Helsinki, Finland. [143]Department of Psychiatry and Psychotherapy, University Medical Center Goettingen, Goettingen, Germany. [144]German Center for Neurodegenerative Diseases (DZNE), Goettingen, Germany. [145]Neurosciences and Signaling Group, Institute of Biomedicine (iBiMED), Department of Medical Sciences, University of Aveiro, Aveiro, Portugal. [146]Department of Psychiatry and Psychotherapy, Universitätsklinikum Erlangen, Friedrich-Alexander Universität Erlangen-Nürnberg, Erlangen, Germany. [147]Department of Population & Quantitative Health Sciences, Case Western Reserve University, Cleveland, OH, USA. [148]Cleveland Institute for Computational Biology, Case Western Reserve University, Cleveland, OH, USA. [149]Translational Neuroscience Laboratory, McLean Hospital, Harvard Medical School, Belmont, MA, USA. [150]Department of Geriatric Psychiatry, University Hospital of Psychiatry Zürich, Zürich, Switzerland. [151]University of Zürich, Zürich, Switzerland. [152]Old age Psychiatry, University Hospital of Lausanne, Lausanne, Switzerland. [153]Department of Psychiatry, Psychosomatics and Psychotherapy, Center of Mental Health, University Hospital, Wuerzburg, Germany. [154]Institute for Stroke and Dementia Research, Klinikum der Universität München, Ludwig-Maximilians-Universität LMU, Munich, Germany. [155]German Center for Neurodegenerative Diseases (DZNE), Munich, Germany. [156]Schools of Life Sciences and Medicine, University of Nottingham, Nottingham, UK. [157]Department of Public Health and Caring Sciences/Geriatrics, Uppsala, Sweden. [158]Institute of Medical Biometry, Informatics and Epidemiology, University Hospital of Bonn, Bonn, Germany. [159]Departments of Medicine (Biomedical Genetics), Neurology, Ophthalmology, Epidemiology, and Biostatistics, Boston University Schools of Medicine and Public Health, Boston, MA, USA. [160]Centre for Memory Disturbances, Lab of Clinical Neurochemistry, Section of Neurology, University of Perugia, Perugia, Italy. [161]Department of Geriatric Psychiatry, Central Institute for Mental Health Mannheim, Medical Faculty Mannheim, University of Heidelberg, Heidelberg, Germany. [162]Department of Epidemiology, Erasmus Medical Center, Rotterdam, The Netherlands. [163]1st Department of Neurology Aristotle University of Thessaloniki, Thessaloniki, Greece. [164]Azienda Ospedaliero-Universitaria, Parma, Italy. [165]Nuffield Department of Clinical Neurosciences, Oxford, UK. [166]Institute of Human Genetics, University of Bonn, School of Medicine & University Hospital Bonn, Bonn, Germany. [167]Munich Cluster for Systems Neurology (SyNergy), Munich, Germany. [168]Department of Primary Medical Care, University Medical Centre Hamburg-Eppendorf, Hamburg, Germany. [169]Laboratory of Cognitive Neuroscience, School of Psychology, Aristotle University of Thessaloniki, Thessaloniki, Greece. [170]Department of Nutrition and Dietetics, Harokopio University, Athens, Greece. [171]Institute of Gerontology and Geriatrics, Department of Medicine, University of Perugia, Perugia, Italy. [172]Institute of Public Health and Clinical Nutrition, University of Eastern Finland, Kuopio, Finland. [173]Neuroepidemiology and Ageing Research Unit, School of Public Health, Imperial College London, London, UK. [174]Stockholms Sjukhem, Research & Development Unit, Stockholm, Sweden. [175]Institute of Biomedicine, University of Eastern Finland, Kuopio, Finland. [176]Brown Foundation Institute of Molecular Medicine, University of Texas Health Sciences Center at Houston, Houston, TX, USA. [177]1st Department of Neurology, Aiginition Hospital, National and Kapodistrian University of Athens, Medical School, Athens, Greece. [178]Taub Institute for Research in Alzheimer's Disease and the Aging Brain, The Gertrude H. Sergievsky Center, Depatment of Neurology, Columbia University, New York, NY, USA. [179]LVR-Hospital Essen, Department of Psychiatry and Psychotherapy, Medical Faculty, University of Duisburg-Essen, Essen, Germany. [180]Department of Psychiatry and Psychotherapy and Experimental and Clinical Research Center (ECRC), Charité-Universitätsmedizin Berlin, Berlin, Germany. [181]German Center for Neurodegenerative Diseases (DZNE), Berlin, Germany. [182]Bristol Medical School (THS), University of Bristol, Southmead Hospital, Bristol, UK. [183]Experimental Neuro-psychobiology Laboratory, Department of Clinical and Behavioral Neurology, IRCCS Santa Lucia Foundation, Rome, Italy. [184]Unit of Neuroscience, DIMEC, University of Parma, Parma, Italy. [185]Athens Association of Alzheimer's disease and Related Disorders, Athens, Greece. [186]Institute of Medical Genetics and Pathology, University Hospital Basel, Basel, Switzerland. [187]Department of Psychiatry, Social Medicine Center East- Donauspital, Vienna, Austria. [188]Center of Mental Health, Clinic and Policlinic of Psychiatry, Psychosomatics and Psychotherapy, University Hospital of Würzburg, Würzburg, Germany. [189]Translational and Clincial Research Institute, Newcastle University, Newcastle upon Tyne, UK. [190]Campus for Ageing anf Vitality, Newcastle upon Tyne, UK. [191]Taub Institute on Alzheimer's Disease and the Aging Brain, Department of Neurology, Columbia University, New York, NY, USA. [192]Gertrude H. Sergievsky Center, Columbia University, New York, NY, USA. [193]Department of Neurology, Columbia University, New York, NY, USA. [194]Laboratory for Cognitive Neurology, Department of Neurosciences, University of Leuven, Leuven, Belgium. [195]Neurology Department, University Hospitals Leuven, Leuven, Belgium. [196]Center for Neurosciences, Vrije Universiteit Brussel (VUB), Brussels, Belgium. [197]Reference Center for Biological Markers of Dementia (BIODEM), University of Antwerp, Antwerp, Belgium. [198]Institute Born-Bunge, University of Antwerp, Antwerp, Belgium. [199]Department of Neurology, VUB University Hospital Brussels (UZ Brussel), Brussels, Belgium. [200]Institute of Clinical Medicine, Internal

Medicine, University of Eastern Finland, Kuopio, Finland. [201]Department of Neurology and Clinical Neurophysiology, University Hospital of Trondheim, Trondheim, Norway. [202]Department of Neuromedicine and Movement Science, Faculty of Medicine and Health Sciences, Norwegian University of Science and Technology, Trondheim, Norway. [203]Department of Laboratory Diagnostics, III Laboratory of Analysis, Brescia Hospital, Brescia, Italy. [204]Alzheimer Research Center & Memory Clinic, Andalusian Institute for Neuroscience, Málaga, Spain. [205]MRC Prion Unit at UCL, Institute of Prion Diseases, London, UK. [206]Department of Medical Genetics, Oslo University Hospital, Oslo, Norway. [207]NORMENT, Department of Clinical Science, University of Bergen, Bergen, Norway. [208]Institute of Social Medicine, Occupational Health and Public Health, University of Leipzig, Leipzig, Germany. [209]Department of Neurology/CIEN Foundation/Queen Sofia Foundation Alzheimer Center, Madrid, Spain. [210]Azienda Ospedaliero-Universitaria Careggi Largo Brambilla, Florence, Italy. [211]UKDRI Cardiff, Cardiff University, Cardiff, UK. [212]Unitat de Genètica Molecular, Institut de Biomedicina de València-CSIC, Valencia, Spain. [213]Unidad Mixta de Neurologia Genètica, Instituto de Investigación Sanitaria La Fe, Valencia, Spain. [214]Department of Neurosciences, Faculty of Medicine and Nursery, University of the Basque Country, San Sebastián, Spain. [215]Centro de Biología Molecular Severo Ochoa (UAM-CSIC), Madrid, Spain. [216]Instituto de Investigacion Sanitaria 'Hospital la Paz' (IdiPaz), Madrid, Spain. [217]Departamento de Especialidades Quirúrgicas, Bioquímica e Inmunología. Facultad de Medicina, Universidad de Málaga, Málaga, Spain. [218]Fundación Pública Galega de Medicina Xenómica-CIBERER-IDIS, Santiago de Compostela, Spain. [219]Framingham Heart Study, Framingham, MA, USA. [220]Nuffield Department of Population Health, University of Oxford, Oxford, UK. [221]Centre for Healthy Brain Ageing (CHeBA), School of Psychiatry, Faculty of Medicine, University of New South Wales, Sydney, NSW, Australia. [222]Neuroscience Research Australia, Sydney, NSW, Australia. [223]Department of Sociology, VU University, Amsterdam, The Netherlands. [224]NORMENT Centre, Institute of Clinical Medicine, University of Oslo, Oslo, Norway. [225]Division of Mental Health and Addiction, Oslo University Hospital, Oslo, Norway. [226]Department of Psychiatry, Glenn Biggs Institute for Alzheimer's and Neurodegenerative Diseases, San Antonio, TX, USA. [306]These authors contributed equally: Itziar de Rojas, Sonia Moreno-Grau, Niccolo Tesi, Benjamin Grenier-Boley, Victor Andrade, Iris E. Jansen. [307]These authors jointly supervised this work: Jordi Clarimón, Mercè Boada, Wiesje M. van der Flier, Alfredo Ramirez, Jean-Charles Lambert, Sven J. van der Lee, Agustín Ruiz. *Lists of authors and their affiliations appears at the end of the paper. ✉email: s.j.vanderlee@amsterdamumc.nl; aruiz@fundacioace.org

## EADB contributors

A. David Smith[227,228], Aldo Saenz[229], Alessandra Bizzarro[230], Alessandra Lauria[230], Alessandro Vacca[137], Alina Solomon[113,131], Anna Anastasiou[163], Anna Richardson[231,232], Anne Boland[233], Anne Koivisto[234,235], Antonio Daniele[236], Antonio Greco[237], Arnaoutoglou Marianthi[163], Bernadette McGuinness[238], Bertrand Fin[233], Camilla Ferrari[66], Carlo Custodero[239], Carlo Ferrarese[100,134], Carlos Ingino[240,241], Carlos Mangone[240,242], Carlos Reyes Toso[240], Carmen Martínez[26,243], Carolina Cuesta[244,245], Carolina Muchnik[240], Catharine Joachim[246], Cecilia Ortiz[247], Céline Besse[233], Charlotte Johansson[41,42], Chiara Paola Zoia[100], Christoph Laske[248,249], Costas Anastasiou[170], Dana Lis Palacio[250,251], Daniel G. Politis[244,245,252], Daniel Janowitz[253], David Craig[238], David M. Mann[254], David Neary[231], Deckert Jürgen[153], Delphine Daian[233], Diyana Belezhanska[255], Eduardo Kohler[256], Eduardo M. Castaño[257,258], Effrosyni Koutsouraki[163], Elena Chipi[160], Ellen De Roeck[196], Emanuele Costantini[236], Emma R. L. C. Vardy[259], Fabrizio Piras[120], Fausto Roveta[137], Federica Piras[120], Federico Ariel Prestia[245,257], Francesca Assogna[120], Francesca Salani[183], Gessica Sala[100], Giordano Lacidogna[236], Gisela Novack[257,258], Gordon Wilcock[260], Håkan Thonberg[41,42], Heike Kölsch[261], Heike Weber[153], Henning Boecker[54,262], Ignacio Etchepareborda[250,251], Irene Piaceri[66], Jaakko Tuomilehto[142,263,264,265], Jaana Lindström[142], Jan Laczo[139,140], Janet Johnston[238], Jean-François Deleuze[233], Jenny Harris[231], Jonathan M. Schott[266], Josef Priller[181,267], Juan Ignacio Bacha[247], Julie Snowden[231,232], Julieta Lisso[268], Kalina Yonkova Mihova[269], Latchezar Traykov[255], Laura Morelli[257,258], Luis Ignacio Brusco[240,241], Malik Rainer[253], Mari Takalo[175], Maria Bjerke[196,270], Maria Del Zompo[47,271], Maria Serpente[80], Mariana Sanchez Abalos (ORCID)[272], Mario Rios[229], Markku Peltonen[142], Martin J. Herrman[153], Mary H. Kosmidis[169], Matias Kohler[250,256], Matias Rojo[240], Matthew Jones[231,273], Michela Orsini[236], Nancy Medel[268], Natividad Olivar[240], Nick C. Fox[266,274], Nicola Salvadori[160], Nigel M. Hooper[232], Pablo Galeano[244,257,258], Patricia Solis[268], Patrizia Bastiani[171], Patrizia Mecocci[171], Peter Passmore[238], Reinhard Heun[261], Riitta Antikainen[275,276,277], Robert Olaso[233], Robert Perneczky[167,278,279,280], Sandra Germani[240], Sara López-García[2,39], Seth Love[182,281], Shima Mehrabian[255], Silvia Bagnoli[66], Silvia Kochen[268], Simona Andreoni[100], Stefan Teipel[282,283], Stephen Todd[238], Stuart Pickering-Brown[254], Teemu Natunen[175], Thomas Tegos[163], Tiina Laatikainen[172,284,285], Timo Strandberg[275,286], Tuomo M. Polvikoski[189,190], Vaclav Matoska[287], Valentina Ciullo[120], Valeria Cores[244,245,252], Vincenzo Solfrizzi[239], Viviana Lisetti[160] & Zulma Sevillano[268]

[227]University of Oxford (OPTIMA), Oxford, UK. [228]OPTIMA, Department of Pharmacology, University of Oxford, Oxford, UK. [229]Dirección de Atención de Adultos Mayores del Min., Salud Desarrollo Social y Deportes de la Pcia. de Mendoza, Mendoza, Argentina. [230]Geriatrics Unit Fondazione Policlinico A. Gemelli IRCCS, Rome, Italy. [231]Cerebral Function Unit, Greater Manchester Neurosciences Centre, Salford Royal Hospital, Salford, UK. [232]Division of Neuroscience and Experimental Psychology, School of Biological Sciences, Faculty of Biology, Medicine and Health, University of Manchester, Manchester, UK. [233]Centre National de Recherche en Génomique Humaine (CNRGH), Institut de Biologie François Jacob, CEA, Université Paris-Saclay, F-91057 Evry, France. [234]Department of Neurology, Kuopio University Hospital, Kuopio, Finland. [235]Insitute of Clinical Medicine-Neurology, University of Eastenr Finland, Kuopio, Finland. [236]Institute of Neurology, Catholic University of Sacred Heart, Fondazione Policlinico Universitario A. Gemelli IRCCS, Rome, Italy. [237]Research Laboratory, Complex Structure of Geriatrics, San Giovanni Rotondo, Fondazione IRCCS Casa Sollievo della Sofferenza, Foggia, Italy. [238]Centre for Public Health, School of Medicine, Queen's University Belfast, Belfast, UK. [239]Clinica Medica "Frugoni" and Geriatric Medicine-Memory Unit, University of Bari Aldo Moro, Bari, Italy. [240]CENECON-FMED-UBA, Buenos Aires, Argentina. [241]ENERI, Vienna, Austria. [242]Htal Santojani, Buenos Aires, Argentina. [243]Servicio de Neurología, Hospital de Cabueñes, Gijón, Spain. [244]UBA, Buenos Aires, Argentina. [245]HIGA Eva Perón, Billinghurst, Argentina. [246]University of Oxford, Oxford, UK. [247]Neurología Clinica, Madrid, Spain. [248]German Center for Neurodegenerative Diseases (DZNE), Tübingen, Germany. [249]Section for Dementia Research, Hertie Institute for Clinical Brain Research and Department of Psychiatry and Psychotherapy, University of Tübingen, Tübingen, Germany. [250]Hospital Dr. Lucio Molas, Santa Rosa, Argentina. [251]Fundacion Ayuda Enfermo Renal y Alta Complejidad (FERNAC), Santa Rosa, Argentina. [252]CONICET, Buenos Aires, Argentina. [253]Institute for Stroke and Dementia Research (ISD), University Hospital, Ludwig-Maximilian University Munich, Munich, Germany. [254]Faculty of Medical and Human Sciences, Institute of Brain, Behaviour and Mental Health, Manchester, UK. [255]Clinic of Neurology UH 'Alexandrovska', Medical University—Sofia, Sofia, Bulgaria. [256]Fundacion Sinapsis, Santa Rosa, Argentina. [257]Laboratory of Brain Aging and Neurodegeneration- FIL, Buenos Aires, Argentina. [258]IIBBA (CONICET), Buenos Aires, Argentina. [259]Salford Royal NHS Foundation Trust, Salford, England. [260]Nuffield Department of Clinical Neurosciences, University of Oxford, Oxford, UK. [261]Department of Psychiatry, University of Bonn, Bonn, Germany. [262]Department of Radiology, University Hospital Bonn, Bonn, Germany. [263]Department of Public Health, University of Helsinki, Helsinki, Finland. [264]National School of Public Health, Madrid, Spain. [265]South Ostrobothnia Central Hospital, Seinäjoki, Finland. [266]Dementia Research Centre, Department of Neurodegenerative Disease, UCL Queen Square Institute of Neurology, University College London, London, UK. [267]Department of Neuropsychiatry, Charité, Berlin, Germany. [268]ENYS (Estudio en Neurociencias y Sistemas Complejos) CONICET-Hospital El Cruce 'Nestor Kirchner'-UNAJ, Buenos Aires, Argentina. [269]Molecular Medicine Center, Department of Medical chemistry and biochemistry, Medical University of Sofia, Sofia, Bulgaria. [270]Laboratory of Neurochemistry, UZ Brussel, Brussel, Belgium. [271]Unit of Clinical Pharmacology, Teaching Hospital of Cagliari, AOUCA, Cagliari, Italy. [272]Programa del Adulto Mayor-Min, Salud de la Pcia. de Jujuy, San Salvador de Jujuy, Argentina. [273]Division of Neuroscience and Experimental Psychology, University of Manchester, Manchester, UK. [274]UK Dementia Research Institute at UCL, UCL Queen Square Institute of Neurology, University College London, London, UK. [275]Center for Life Course Health Research, University of Oulu, Oulu, Finland. [276]Medical Research Center Oulu, Oulu University Hospital, Oulu, Finland. [277]Oulu City Hospital, Oulu, Finland. [278]German Center for Neurodegenerative Diseases (DZNE, Munich), Munich, Germany. [279]Department of Psychiatry and Psychotherapy, University Hospital LMU Munich, Munich, Germany. [280]Ageing Epidemiology Research Unit (AGE), School of Public Health, Imperial College London, London, UK. [281]University of Bristol Institute of Clinical Neurosciences, School of Clinical Sciences, Frenchay Hospital, Bristol, UK. [282]German Center for Neurodegenerative Diseases (DZNE), Rostock, Germany. [283]Department of Psychosomatic Medicine, Rostock University Medical Center, Rostock, Germany. [284]Finnish Institute for Health and Welfare, Helsinki, Finland. [285]Joint Municipal Authority for North Karelia Social and Health Services (Siun Sote), Joensuu, Finland. [286]Helsinki University Hospital, University of Helsinki, Helsinki, Finland. [287]Department of Clinical Biochemistry, Hematology and Immunology, Na Homolce Hospital, Prague, Czech Republic.

## The GR@ACE study group

C. Abdelnour[1,2], N. Aguilera[1], E. Alarcon[1,31], M. Alegret[1,2], A. Benaque[1], M. Boada[1,2], M. Buendia[1], P. Cañabate[1,2], A. Carracedo[18,218], A. Corbatón-Anchuelo[6], I. de Rojas[1], S. Diego[1], A. Espinosa[1,2], A. Gailhajenet[1], P. García-González[1], S. Gil[1], M. Guitart[1], A. González-Pérez[19], I. Hernández[1,2], M. Ibarria[1], A. Lafuente[1], J. Macias[23], O. Maroñas[18], E. Martín[1], M. T. Martínez[62], M. Marquié[1], A. Mauleón[1], L. Montrreal[1], S. Moreno-Grau[1,2], M. Moreno[1], A. Orellana[1], G. Ortega[1,2], A. Pancho[1], E. Pelejá[1], A. Pérez-Cordon[1], J. A. Pineda[23], S. Preckler[1], I. Quintela[31], L. M. Real[8,31], M. Rosende-Roca[1], A. Ruiz[1,2], M. E. Sáez[19], A. Sanabria[1,2], M. Serrano-Rios[61], O. Sotolongo-Grau[1], L. Tárraga[1,2], S. Valero[1,2] & L. Vargas[1]

## DEGESCO consortium

A. D. Adarmes-Gómez[2,22], E. Alarcón-Martín[1,31], M. D. Alonso[288], I. Álvarez[37,38], V. Álvarez[26,138], G. Amer-Ferrer[289], M. Antequera[13], C. Antúnez[13], M. Baquero[36], M. Bernal[22], R. Blesa[2,24], M. Boada[1,2], D. Buiza-Rueda[2,22], M. J. Bullido[2,215,216], J. A. Burguera[36], M. Calero[2,21,40], F. Carrillo[2,22], M. Carrión-Claro[2,22], M. J. Casajeros[290], J. Clarimón[2,24], J. M. Cruz-Gamero[31], M. M. de Pancorbo[291], I. de Rojas[1,2], T. del Ser[216], M. Diez-Fairen[37,38], R. Escuela[2,22], L. Garrote-Espina[2,22], J. Fortea[2,24], E. Franco-Macías[22], A. Frank-García[2,29,216], J. M. García-Alberca[204], S. Garcia Madrona[40], G. Garcia-Ribas[40], P. Gómez-Garre[2,22], I. Hernández[1,2], S. Hevilla[204], S. Jesús[2,22], M. A. Labrador Espinosa[2,22], C. Lage[2,39], A. Legaz[13], A. Lleó[2,24], A. Lopez de Munain[32,33], S. López-García[2,39], D. Macias-García[2,22], S. Manzanares[289,292], M. Marín[22], J. Marín-Muñoz[13], T. Marín[204], M. Marquié[1,2], A. Martín Montes[2,29,215], B. Martínez[13], C. Martínez[26,243],

V. Martínez[13], P. Martínez-Lage Álvarez[293], M. Medina[2,216], M. Mendioroz Iriarte[294], M. Menéndez-González[25,26], P. Mir[2,22], J. L. Molinuevo[295], L. Montrreal[1], A. Orellana[1], P. Pastor[37,38], J. Pérez Tur[2,212,213], T. Periñán-Tocino[2,22], R. Pineda-Sanchez[2,22], G. Piñol-Ripoll[2,34,35], A. Rábano[2,21,46], D. Real de Asúa[296], S. Rodrigo[22], E. Rodríguez-Rodríguez[2,39], J. L. Royo[31], A. Ruiz[1,2], R. Sanchez del Valle Díaz[14], P. Sánchez-Juan[2,39], I. Sastre[2,215], O. Sotolongo-Grau[1], S. Valero[1,2], M. P. Vicente[13], R. Vigo-Ortega[2,22] & L. Vivancos[13]

[288]Servei de Neurologia, Hospital Clínic Universitari de València, València, Spain. [289]Department of Neurology, Hospital Universitario Son Espases, Palma, Spain. [290]Hospital Universitario Ramón y Cajal, Madrid, Spain. [291]BIOMICs, País Vasco, Centro de Investigación Lascaray, Universidad del País Vasco UPV/EHU, Leioa, Spain. [292]Fundación para la Formación e Investigación Sanitarias de la Región de Murcia, El Palmar, Spain. [293]Fundación CITA-alzheimer, Centro de Investigació'n y Terapias Avanzadas, San Sebastián, Spain. [294]Navarrabiomed, Pamplona, Spain. [295]Barcelona beta Brain Research Center–Fundació Pasqual Maragall, Barcelona, Spain. [296]Hospital Universitario La Princesa, Madrid, Spain.

## IGAP (ADGC, CHARGE, EADI, GERAD)

C. Macleod[297], C. McCracken[298], Carol Brayne[299], Catherine Bresner[17], Detelina Grozeva[17], Eftychia Bellou[17], Ewen W. Sommerville[17], F. Matthews[300], Ganna Leonenko[17], Georgina Menzies[50], Gill Windle[297], Janet Harwood[17], Judith Phillips[301], K. Bennett[302], Lauren Luckuck[17], Linda Clare[303], Robert Woods[304], Salha Saad[17] & Vanessa Burholt[305]

[297]School of Health Sciences, Bangor University, Bangor, UK. [298]Institute of Psychology, Health and Society, University of Liverpool, Liverpool, England. [299]School of Clinical Medicine, Cambridge Institute of Public Health, University of Cambridge, Cambridge, England. [300]Faculty of Medicine, Institute of Health and Society, University of Newcastle, Callaghan, NSW, Australia. [301]Dementia Studies, University of Stirling, Stirling, Scotland. [302]School of Psychology, University of Liverpool, Liverpool, England. [303]School of Psychology, Centre for Research in Ageing and Cognitive Health (REACH), University of Exeter, Exeter, England. [304]Dementia Services Development Centre, Bangor University, Wales, UK. [305]Swansea University, School of Human Sciences, Centre for Innovative Ageing, Swansea, UK.

## PGC-ALZ consortia

Iris E. Jansen[3,9], Nancy L. Pedersen[10], Arvid Rongve[62], Chandra A. Reynolds ⓘ[79], Dag Aarsland ⓘ[89,90], Geir Selbæk ⓘ[116,117,118], Ida K. Karlsson[10,133], Ingvild Saltvedt ⓘ[135,136], Sara Hägg ⓘ[10], Sigrid Botne Sando[201,202], Srdjan Djurovic ⓘ[206,207], Ole A. Andreassen ⓘ[224,225], Danielle Posthuma[9]

