## [Peer Review File · Nature Communications]

Reviewers' Comments:

Reviewer #1:

Remarks to the Author:

I have previously reviewed the submission for Nature Genetics, hence I will not repeat my previous review but address the author reply.

Overall, the authors have addressed some of my concerns and clarified some issues, mainly around the PRS analysis. At the same time, the authors seem resistant to meaningfully changing any of their analyses to increase the statistical power of their study and seem content to simply reword their aims instead. Even just changing figures to increase the clarity of the study design was not considered.

Many of the modifications or clarifications are quite cursory and brief, and some are only presented here in the reply and not in the main text, or vice-versa. Often I am left trying to guess at the authors' intentions.

Main comments:

1. LDSC analysis: this result seems fine and suggests low population stratification. But please give a bit more information in the Methods and cite the LDSC paper.
2. Figure 1 and workflow. I thank the authors for clarifying but Figure 1 has not been modified at all from the previous submission and I still find it confusing; I suspect many readers will as well. For example, it's very difficult to follow why there are different subsets of GR@CE in Figure 1 (with different sample sizes) and how they relate to each other.
3. PRS construction: the authors state that the objectives have been rewritten, but it is not clear exactly what they are referring to, and I am left guessing. (I could not find the details in reply 2 to reviewer 1).
4. Survival analysis: my earlier comments still stand and have not been fully addressed.
 - (i) The Cox regression can be performed with case-only (using age of onset and no censoring), and will likely provide substantially increased power over the current binning strategy, while allowing adjustment for covariates like APOE status and sex (as well as testing for interactions). There are other regression methods that can be used as well. The binning approach is very underpowered and limits the conclusions of the study.
 - (ii) The clarification that the GR@ACE controls are much younger than cases:

this means that some of the controls might still develop AD later in life, and will eventually be cases. This fact does not completely invalidate the analysis assuming that the proportion of 'eventual AD' in the controls is low enough. But this point needs to be acknowledged somewhere in the Discussion.

5. PRS independent of APOE status: I assume that the table shows odds ratios for the PRS within each APOE group; this is not specified in the reply. Please add this table to the manuscript (beyond the one sentence in Figure 4).

6. GTEx analysis: As with my previous comment, there is no information in the manuscript on this analysis. Whether the results are based on public databases or not, it needs describing: what the analysis was, which data was used, etc.

7. Limitation of European ancestry analysis: This comment has been addressed. I suggest that the Discussion is a better place to explain this issue than the Method section.

Minor comments:

* The following Abstract sentence is quite unclear: "Because of this study the underlying pathological mechanisms of the common risk allele in APP can be studied to refine the amyloid cascade".

Reviewer #2:

Remarks to the Author:

The authors have been responsive to all of my comments. I have one minor suggestion, it would be nice to compare the p-value thresholding used in their PRS analysis to the paper published by Peter Visscher Oligogenic paper. Are the authors able to replicate the findings from the oligogenic paper?

Referees' comments:

Referee #1 (Remarks to the Author):

I have previously reviewed the submission for Nature Genetics, hence I will not repeat my previous review but address the author reply.

Overall, the authors have addressed some of my concerns and clarified some issues, mainly around the PRS analysis. At the same time, the authors seem resistant to meaningfully changing any of their analyses to increase the statistical power of their study and seem content to simply reword their aims instead. Even just changing figures to increase the clarity of the study design was not considered.

Many of the modifications or clarifications are quite cursory and brief, and some are only presented here in the reply and not in the main text, or vice-versa. Often I am left trying to guess at the authors' intentions.

Main comments:

1. LDSC analysis: this result seems fine and suggests low population stratification. But please give a bit more information in the Methods and cite the LDSC paper.

Answer: In accordance with reviewer, we have now added the information about the LDSC in Methods section: *"Polygenicity and confounding biases, such as cryptic relatedness and population stratification, can yield an inflated distribution of test statistics in GWAS. To distinguish between inflation from a true polygenic signal and bias we quantified the contribution of each by examining the relationship between test statistics and linkage disequilibrium (LD) using the LD Score regression intercept (LDSC software³³)."*

2. Figure 1 and workflow. I thank the authors for clarifying but Figure 1 has not been modified at all from the previous submission and I still find it confusing; I suspect many readers will as well. For example, it's very difficult to follow why there are different subsets of GR@CE in Figure 1 (with different sample sizes) and how they relate to each other.

Answer: We acknowledge it is confusing. GR@ACE includes on average 1500-2000 samples per year, in the time between finishing the discovery, additional samples were genotyped. We added these as replication. State it as GR@ACE-replication.

The workflow in Figure 1 has been slightly modified for better understanding. Also we added some comments in the figure legend. *"Fig. 1. (...)The genome-wide significant signals found in meta-GWAS were used to perform a Polygenic Risk Score in a clinical and pathological AD dataset. See supplementary methods to more information about the cohorts included and methods to the PRS generation. ^aExtended dataset (S.Moreno-Grau et al. 2019), ^bStage! + Stage!! (Kunkle et al. 2019), ^cBy proxy AD: Meta-analysis of maternal and paternal history of dementia (Marioni et al. 2018), ^dExtra and independent GR@ACE dataset incorporated only for replication purposes, ^ePathologically confirmed AD cases, ^fAD cases diagnosed based on clinical criteria, ^gControls participants aged 55 years and younger. N = Total of individuals within specified data."*

3. PRS construction: the authors state that the objectives have been rewritten, but it is not clear exactly what they are referring to, and I am left guessing. (I could not find the details in reply 2 to reviewer 1).

Answer: The discrimination and optimization of PRS is important. However, in this work, the objective of genetic PRS was to add to the existing data that the PRS can identify subjects at highest risk of AD. First, we performed a validation study to show that PRS (including the new SNP list) is a strong and independent predictor of important determinants of AD (age at onset, gender, PCs, APOE, or diagnosis accuracy (pathological versus clinical series). Then we performed the extensive stratification of risk by the PRS in the GR@ACE dataset, which has not been shown previously. Lastly, we'd like to emphasize that this validation dataset and the large GR@ACE dataset are completely independent from other published cohorts and therefore our results strongly reinforce the evidence for the effect of the PRS in AD.

The PRS study should thus be considered only as a proof of principle for future prospective studies that will follow. Furthermore, we know that our PRS cannot be considered definitive because it will be refreshed and improved periodically by introducing new hits yet to be discovered and added to the model in the future.

We hope that the rewriting of the PRS goals will make it clear that there are many ways to construct the PRS and we did not aim to identify the 'best' PRS, neither to make the comparison to other PRS or polygenic hazard scores (Qian Zhang et al., *Nature Communications* 2020).

4. Survival analysis: my earlier comments still stand and have not been fully addressed.

(i) The Cox regression can be performed with case-only (using age of onset and no censoring), and will likely provide substantially increased power over the current binning strategy, while allowing adjustment for covariates like APOE status and sex (as well as testing for interactions). There are other regression methods that can be used as well. The binning approach is very underpowered and limits the conclusions of the study.

(ii) The clarification that the GR@ACE controls are much younger than cases: this means that some of the controls might still develop AD later in life, and will eventually be cases. This fact does not completely invalidate the analysis assuming that the proportion of 'eventual AD' in the controls is low enough. But this point needs to be acknowledged somewhere in the Discussion.

Answer: (i) We implemented the Cox regression model case-only as a request of the reviewer and we report it in the methods and results section.

Methods: "We implemented a Cox regression model on the GR@ACE/DEGESCO dataset case-only adjusted for covariates as APOE group, the interaction between the PRS and APOE and four population ancestry components. All analyses were done in R (v3.4.2)."

Results: "The Cox regression also showed an impact of APOE on AAO, mainly on APOE $\epsilon 4\epsilon 4$ (significant APOE-PRS interaction ($p = 0.021$), Fig. 5d)."

(ii) In accordance with reviewer's suggestions, the limitations related to the use of younger controls have been reinforced in the discussion section

5. PRS independent of APOE status: I assume that the table shows odds ratios for the PRS within each APOE group; this is not specified in the reply. Please add this table to the manuscript (beyond the one sentence in Figure 4).

Answer: The table was added to the supplementary tables also pointing to the results of Figure 4 in the Results section.

Supplementary Table 7. Association of the continuous AD-PRS within each APOE groups.

APOE groups	P-value	OR [CI 95%]
€2€2/€2€3	1.9×10^{-05}	1.33[1.17-1.51]
€3€3	9.2×10^{-21}	1.25[1.19-1.31]
€2€4/€3€4	1.9×10^{-13}	1.32[1.23-1.42]
€4€4	0.05	1.29[1.00-1.68]

6. GTEx analysis: As with my previous comment, there is no information in the manuscript on this analysis. Whether the results are based on public databases or not, it needs describing: what the analysis was, which data was used, etc.

Answer: Following the reviewer's advice, we added some information about GTEx analysis in the supplementary information.

"Expression data in public databases.

We use GTEx Portal19 and BrainSeq23 eQTL Browser to identify expression quantitative trait loci (eQTLs) in the genome wide significant loci in this study. For GTEx we download the multi-tissue eQTLcomparison (data Source: GTEx Analysis Release V8 (dbGaP Accession phs000424.v8.p2), see details in <https://gtexportal.org/home/>.

The BrainSeq is based on the dorsolateral prefrontal cortex (DLPFC) polyA+ RNA-seq on 738 subjects spanning the lifespan and three main psychiatric diagnostic groups (Schizophrenia, Major Depression Disorder, and Bipolar Disorder). BrainSeq identified eQTLs in the DLPFC using RNA sequencing (RNA-seq) and genotype data. The eQTL modeling tested for additive genetic effects on expression while adjusting for sex, ancestry (multidimensional scaling components), and expression heterogeneity (principal components). Significant eQTLs were those SNP-feature pairs with a false discovery rate (FDR) less than 1%. The "DLPFC - All" database was used. For more details, see <http://eqtl.brainseq.org/phase1/eqtl/>."

7. Limitation of European ancestry analysis: This comment has been addressed. I suggest that the Discussion is a better place to explain this issue than the Method section.

Answer: Following the reviewer's advice, the sentence has been moved to discussion.

Minor comments:

* The following Abstract sentence is quite unclear: "Because of this study the underlying pathological mechanisms of the common risk allele in APP can be studied to refine the amyloid cascade".

Answer: We are trying to convey is that due to the findings of this study regarding the common protective allele in APP, may help to refine the amyloid cascade hypothesis for Alzheimer's disease. We have rewritten the sentence to clarify it.

Reviewer #2 (Remarks to the Author):

The authors have been responsive to all of my comments. I have one minor suggestion, it would be nice to compare the p-value thresholding used in their PRS analysis to the paper published by Peter Visscher Oligogenic paper. Are the authors able to replicate the findings from the oligogenic paper?

Answer: The Oligogenic paper show that the discriminative ability of GRS in LOAD prediction is maximized when selecting a small number of SNPs. Both simulation results and direct estimation indicate that the number of causal common SNPs for LOAD may be less than 100, suggesting LOAD is more oligogenic than polygenic.

We showed that adding variants below the GWS threshold ($>1 \times 10^{-7}$, $>1 \times 10^{-6}$, $>1 \times 10^{-5}$, $>1 \times 10^{-4}$, $>1 \times 10^{-3}$, $>1 \times 10^{-2}$) did not lead to a more significant association of the PRS with AD (Fig. 4a, extracted below). So yes, we in fact confirm it and we also integrated the mentioned paper as a reference of our manuscript.

Fig. 4. Polygenic Risk Scores for AD. a, The 39-SNP PRS association with clinical (OR = 1.38, per 1-SD increase in the PRS, 95% CI [1.21–1.58], $p = 1.5 \times 10^{-6}$) and pathologically confirmed AD cases (OR = 1.30, 95% CI [1.18–1.44], $p = 1.1 \times 10^{-7}$).

Reviewers' Comments:

Reviewer #1:

Remarks to the Author:

The authors have addressed most of my previous comments.

One remaining issue is the survival analysis (Cox regression). It's not clear what was actually done here, because there is very little detail, and it's not clear how to interpret the results of Fig 5d in the context of Cox regression.

The standard results from a Cox regression would be the hazard ratios for the PRS, APOE status, other covariates, and/or interactions. Fig 5d actually looks like Kaplan-Meier curves instead (not the same thing, although useful in its own right).

Also, the notation for the y-axis doesn't make sense, how does the probability of being a case _go down_ with age? I suspect this is plotting the survival curve, i.e., $1 - \text{risk}$.

Referee comments:

Referee #1 (Remarks to the Author):

The authors have addressed most of my previous comments.

One remaining issue is the survival analysis (Cox regression). It's not clear what was actually done here, because there is very little detail, and it's not clear how to interpret the results of Fig 5d in the context of Cox regression. The standard results from a Cox regression would be the hazard ratios for the PRS, APOE status, other covariates, and/or interactions. Fig 5d actually looks like Kaplan-Meier curves instead (not the same thing, although useful in its own right). Also, the notation for the y-axis doesn't make sense, how does the probability of being a case go down with age? I suspect this is plotting the survival curve, i.e., 1 - risk.

Answer: We apologize, for not including sufficient information in the legend. The plot is a visual representation on a cox-model based on cases-only and with age at onset as the time variable. The determinants are the PRS and the APOE categories (APOE $\epsilon 2\epsilon 2/\epsilon 2\epsilon 3$, APOE $\epsilon 3\epsilon 3$, APOE $\epsilon 3\epsilon 4/\epsilon 2\epsilon 4$, APOE $\epsilon 4\epsilon 4$), a PRS*APOE interaction term and population substructure as covariates. We then fixed the covariates to their mean value and the PRS to either -2SD or +2SD. This then corresponds to eight probability curves shown in figure 5D. The curve shows the probability a case in one of the eight groups has developed AD by a certain age (x-axis). One can read from the graph that 50% of APOE $\epsilon 4\epsilon 4$ carriers with high PRS score have developed AD by the age of 74, compared to 78 years of age for APOE $\epsilon 4\epsilon 4$ carriers with a low PRS. More details of the analysis were added.